# Learning to Bid with Unknown Private Values in Budget-Constrained First-Price Auctions

## Abstract

The transition to First-Price Auctions (FPA) in digital advertising has spurred significant research, yet existing work typically assumes access to a valuation oracle, ignoring the reality that values must be inferred from censored data. While Linear Treatment Effect (LTE) models address this by learning value uplift, they have not been adapted to realistic settings with strictly enforced Budget or Return-on-Spend (RoS) constraints. In this work, we propose a unified primal-dual framework for constrained FPAs that jointly learns the latent LTE valuation parameters and the competitor's bid distribution. This simultaneous learning introduces a critical technical challenge: the estimation error is dynamically scaled by the Lagrangian multiplier, potentially leading to unbounded regret. We resolve this by leveraging a strong Slater condition and a novel adaptive burn-in procedure to stabilize the dual variables. Our approach achieves near-optimal regret guarantees, providing the first theoretically grounded solution for constrained bidding with latent valuations.

## 1. Introduction

Digital advertising remains a cornerstone of the modern internet economy, representing a multi-billion dollar market with immense growth potential. Recently, the industry has shifted predominantly from Second-Price Auctions (SPA) to **First-Price Auctions (FPA)** (Bigler, 2019). This transition forces advertisers to rethink bidding strategies, moving from truth-telling to bid shading to mitigate transparency concerns (Despotakis et al., 2021). However, maximizing surplus is rarely unconstrained. In practice, major ad exchanges impose strict financial guardrails. As illustrated by Google Ads (Figure 1), advertisers are typically **required**

to set a budget, while often **optionally** selecting goals like Return on Spend (RoS).

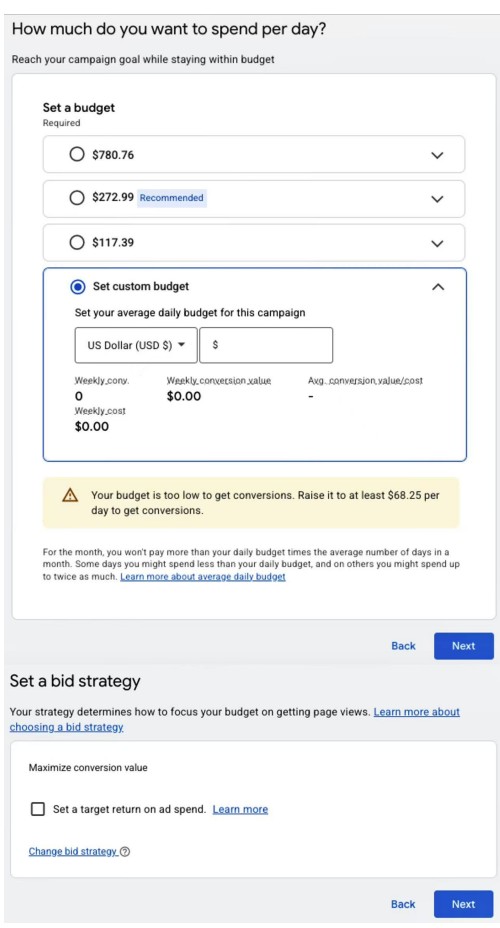

*Figure 1.* Google Ads settings interface for Budget constraints (mandatory) and Target ROAS (optional).

Despite the prevalence of these constraints, a fundamental disconnect remains in the literature regarding the **valuation** of ad impressions. Existing theoretical frameworks in this domain—including works on constrained bidding (Wang et al., 2023; Li et al., 2025; Vijayan et al., 2025)—operates under the assumption that the true value of an impression is revealed to the bidder either prior to the auction or immediately upon winning. This raises a critical question: *Where*

---

[1]Anonymous Institution, Anonymous City, Anonymous Region, Anonymous Country. Correspondence to: Anonymous Author <anon.email@domain.com>.

Preliminary work. Under review by the International Conference on Machine Learning (ICML). Do not distribute.

*do these values come from?*

Directly evaluating the causal value of an impression requires human expertise that is unscalable in real-time bidding. To bridge this gap, we adopt the Linear Treatment Effect (LTE) model, assuming the "uplift" value is generated by a linear model based on observable context vectors. However, in the highly competitive ecosystem of Real-Time Bidding (RTB), the influence of context extends beyond just our own valuation. The context vector $x_t$ summarizes key properties of the ad impression—such as user demographics, device type, and browsing history—and is observable to all bidders with potential interest. For instance, an ad impression targeting a customer from the Upper East Side using an iPhone signals a high Click-Through Rate (CTR) not only to us but to our competitors as well. Consequently, it is natural to assume that the market competition is structurally similar to our own valuation logic.

Therefore, our setting is more specific than standard LTE approaches: we posit that both the uplift value and the opponents' highest bid are linearly generated by the same context vector $x_t$. This joint dependence distinguishes our contribution from prior literature. While previous works like Waisman et al. (2019); Wen et al. (2025) focus on the linear treatment effect on uplift value, and others consider linear models solely for opponent bids (Badanidiyuru et al., 2023). To the best of our knowledge, we provide the first constructive regret analysis for online first-price auctions where both the value uplift and the market price jointly depend on the same underlying context vector.

Therefore, our setting is more specific than standard LTE approaches: we posit that both the uplift value and the opponents' highest bid are linearly generated by the same context vector $x_t$. Assuming linearity with respect to a feature vector does not imply the world is simple, but rather that the feature extractor is competent. This joint dependence distinguishes our contribution from prior literature. While previous works like (Waisman et al., 2019; Wen et al., 2025) focus on the linear treatment effect on uplift value, and others consider linear models solely for opponent bids (Badanidiyuru et al., 2023), we are the first to conduct a systematic study of the setting where both the value uplift and the market price jointly depend on the same underlying context vector.

**Challenges.** Although Wen et al. (2025) explore the joint learning of uplift values and competitor bids, their analysis is confined to **unconstrained** settings and relies on non-constructive existence proofs (Marcus et al., 2015). Generalizing this to constrained settings introduces non-trivial challenges driven by the need for joint estimation. The primary difficulty stems from the coupling between *latent value uncertainty* and *constraint enforcement*. In the **RoS-constrained** regime, this coupling manifests as a haz-ardous interaction term between the Lagrangian multiplier and the estimation confidence radius. Unlike standard settings where multipliers and errors are separable, here, a large multiplier (required for tight constraints) can amplify the estimation error, rendering the regret bound theoretically unbounded. Furthermore, the non-convex nature of first-price auction payments complicates the optimization landscape, preventing direct applications of existing convexity-based analyses.

**Our Contributions.** In this work, we propose a unified framework for Online First-Price Auctions with Linear Treatment Effects, progressing from unconstrained to Budget-constrained, and finally RoS-constrained settings. Our main contributions are:

- **Unified Modeling Framework:** We establish a framework for online first-price auctions where both the treatment effect and the highest competing bid are linearly dependent on the context. This captures the realistic dependency between user features and auction dynamics, allowing us to extend the analysis from unconstrained bidding to complex global constraints within a single Primal-Dual architecture.

- **Optimal Regret Rates in $T$:** We provide rigorous regret analysis for three settings: unconstrained, hard budget constraint, and expected RoS constraint. We show that our algorithms achieve a cumulative regret of $\tilde{O}(d\sqrt{T})$. Since the lower bound for the unconstrained setting scales with $\Omega(\sqrt{T})$, our results are optimal with respect to the time horizon, demonstrating that adhering to global constraints does not degrade the fundamental learning rate.

- **Constructive Estimation and Dual Stability:** We bridge the gap between theory and practice in two aspects. First, we propose a computationally efficient **split-sample estimation procedure** for the competing bid distribution, providing a constructive alternative to the existential guarantees in prior LTE work (Wen et al., 2025). Second, for the RoS setting where the dual domain is unknown, we introduce an **adaptive burn-in phase** to dynamically estimate the Slater constant, ensuring primal-dual convergence without prior knowledge of optimal dual variables.

## 1.1. Related Work

**Online First-price Auctions:** Due to the transition from second-price auctions to first-price auctions by major ad exchanges (e.g., Google), there has been significant interest in how to bid in first-price auctions. From a theoretical perspective, researchers study the equilibrium behavior of bidding strategies (Filos-Ratsikas et al., 2021; Conitzer et al., 2022; Chen & Peng, 2023; Gaitonde et al., 2022; Lucier

et al., 2024). From a learning perspective, researchers focus on designing bidding strategies for a variety of environments (Han et al., 2020; 2025; Zhang et al., 2022; Balseiro & Gur, 2019; Wang et al., 2023; Li et al., 2025; Hu et al., 2025; Vijayan et al., 2025). Among them, Badanidiyuru et al. (2023) is especially related to our work: it considers a setting in which the opponents' highest bid follows a linear model and the value is observed prior to bidding. In contrast, we address a "doubly linear" setting where both the competitive landscape (opponent bids) and the bidder's own uplift value depend jointly on the context vector.

**Online Bidding with Constraints:** In real-world auto-bidding settings, it is especially useful to consider the budget/RoS constraint (Aggarwal et al., 2024). Wang et al. (2023) consider online first-price auctions with budget constraints. Feng et al. (2023) study bidding in second-price auctions under an RoS constraint. Li et al. (2025) propose an algorithm for first-price auctions with an RoS constraint, based on a novel convexification argument. A critical limitation shared by these works (Wang et al., 2023; Feng et al., 2023; Li et al., 2025) is the assumption that the value of an impression is fully observable prior to bidding. While Vijayan et al. (2025) relax this by considering values revealed only upon winning, they still rely on ex-post observability. Our work departs from this paradigm by targeting uplift values that are never directly observable, requiring causal inference techniques. Furthermore, unlike Vijayan et al. (2025), which is restricted to discrete bid sets, our approach extends naturally to continuous bid spaces.

**Online Learning with Constraints:** Online auction with budget/RoS constraint falls within the broader literature on online learning with constraints. The budget constraint is typically handled by Knapsack bandit techniques (Agrawal & Devanur, 2016; Badanidiyuru et al., 2018; Immorlica et al., 2022). Due to the non-packing property and the non-convex nature of the RoS constraint, one typically considers randomized strategies and applies the primal-dual framework Castiglioni et al. (2022a;b). In principle, the results of Castiglioni et al. (2022a) could be used to handle our RoS setting. However, Castiglioni et al. (2022b) needs to invoke a non-convex primal optimizer, such as the one in Suggala & Netrapalli (2020). By leveraging recent convexification insights for first-price auctions (Li et al., 2025), we streamline this process, enabling an efficient deterministic bidding strategy without sacrificing theoretical guarantees.

## 2. Problem Formulation and Main Results

### 2.1. Notations

We use $[n]$ to denote the set $\{1, 2, \ldots, n\}$ and $\mathbb{I}[\cdot]$ to denote the indicator function. Following standard asymptotic notation, we use $O(\cdot)$ and $\Omega(\cdot)$ to suppress constant fac-

tors, while $\tilde{O}(\cdot)$ is employed to hide polylogarithmic factors. The notation $\mathbb{E}[\cdot]$ is used to denote the expectation of a random variable. For a positive semi-definite (PSD) matrix $A \in \mathbb{R}^{d \times d}$, we define the weighted (local) norm as $\|x\|_A = \sqrt{x^\top A x}$. We use $\mathcal{B}$ to denote the continuous bidding space $[0, 1]$ and $\mathcal{B}_K$ to denote its discretized version $\{0, 1/K, 2/K, \ldots, 1\}$.

### 2.2. Problem Formulation

In this work, we consider a single bidder participating in repeated First-Price Auctions (FPA) against a population of other bidders over a time horizon $T$, (optionally) subject to a hard budget constraint $B$ or an expected RoS target. At the beginning of each time step $t \in [T]$, the bidder observes context information $x_t \in \mathbb{R}^d$ (common knowledge to all bidders, potentially empty), drawn i.i.d. from $\mathcal{D}$, satisfying $\|x_t\|_2 \leq 1$. After submitting a bid $b_t \in \mathcal{B} = [0, 1]$, the bidder observes the highest competing bid (HOB) $m_t$ regardless of the outcome (full-information feedback on $m_t$). If $b_t \geq m_t$, the bidder receives an outcome value $v_{t,1} \in [0, 1]$ and pays the bid amount $b_t$. Otherwise, the bidder obtains an outcome value $v_{t,0} \in [0, 1]$ and pays nothing. We use $c_t(b_t) := \mathbb{I}[b_t \geq m_t] \cdot b_t$ to denote the payment function. For the budget constrained and unconstrained settings, the learner's realized reward (utility) is defined as the net uplift value obtained:

$$r_t(b_t) = \mathbb{I}[b_t \geq m_t](v_{t,1} - v_{t,0} - b_t). \qquad (1)$$

While for the RoS constrained setting, the learner's realized reward (utility) is defined as the uplift value obtained:

$$r_t(b_t) = \mathbb{I}[b_t \geq m_t](v_{t,1} - v_{t,0}). \qquad (2)$$

It is common to consider value maximization under RoS constraints (Feng et al., 2023; Li et al., 2025; Vijayan et al., 2025), whereas utility maximization is more suitable for the unconstrained setting, because a value-maximizing bidder can bid the highest possible bid to win every ad impression.

To model the context vector as common knowledge to all bidders, we assume:

$$\mathbb{E}[v_{t,1} - v_{t,0} \mid x_t] = \theta_\star^\top x_t, \qquad m_t = \phi_\star^\top x_t + \xi_t, \quad (3)$$

where $\theta_\star, \phi_\star \in \mathbb{R}^d$ are unknown parameters, and $\xi_t$ are random variables with zero mean. This means all bidders can apply the contextual information of the ad impression to facilitate bidding. We use $F_t$ to denote the CDF of $m_t$.

### 2.3. Budget Constraint and Regret Definition

The bidder operates under a strict total budget $B$. The process must stop once cumulative payments would exceed the budget. Formally, we define the stopping time $\tau$ as the

first round at which the remaining budget is insufficient to cover the bid:

$$\tau = \min \left\{ t \in [T+1] : \sum_{s=1}^{t-1} c_s(b_s) + b_t > B \right\}. \quad (4)$$

If the budget constraint is never violated over the horizon, $\tau = T+1$. For all rounds $t \geq \tau$, the bidder is forced to stop bidding (effectively $b_t = 0$), resulting in zero payment and no treatment effect gain. Due to the global budget constraint, the standard per-round regret is insufficient. Instead, we compare the cumulative reward of the learning algorithm against an optimal stationary policy $\pi^*$. Let $\Pi$ be the set of all stationary policies that map contexts to bids. The optimal benchmark $\pi^*$ is defined as:

$$\pi^* = \arg\max_{\pi \in \Pi} \mathbb{E} \left[ \sum_{t=1}^{T} r_t(\pi(x_t)) \right] \quad (5)$$

subject to $\mathbb{E} \left[ \sum_{t=1}^{T} c_t(\pi(x_t)) \right] \leq B$.

The bidder aims to minimize the cumulative regret, defined as:

$$R_T^{\mathrm{Bgt}} = \mathbb{E} \left[ \sum_{t=1}^{T} r_t(\pi^*(x_t)) \right] - \sum_{t=1}^{\tau-1} r_t(b_t). \quad (6)$$

When the budget is sufficiently large ($B = \infty$) and the stopping time satisfies $\tau = T+1$, we recover the unconstrained regret:

$$R_T^{\mathrm{Unc}} = \mathbb{E} \left[ \sum_{t=1}^{T} r_t(\pi^*(x_t)) \right] - \sum_{t=1}^{T} r_t(b_t). \quad (7)$$

## 2.4. RoS Constraint

Following Li et al. (2025), we consider the expected RoS constraint $\mathbb{E} \left[ \sum_{t=1}^{T} (r_t(b_t) - \rho c_t(b_t)) \right] \geq 0$, which means that spending one dollar yields $\rho$ dollars of uplift in expectation. The optimal benchmark $\pi^*$ is defined as:

$$\pi^* = \arg\max_{\pi \in \Pi} \mathbb{E} \left[ \sum_{t=1}^{T} r_t(\pi(x_t)) \right] \quad (8)$$

subject to $\mathbb{E} \left[ \sum_{t=1}^{T} (r_t(\pi(x_t)) - \rho c_t(\pi(x_t))) \right] \geq 0$. The regret is defined as:

$$R_T^{\mathrm{RoS}} = \mathbb{E} \left[ \sum_{t=1}^{T} r_t(\pi^*(x_t)) \right] - \mathbb{E} \left[ \sum_{t=1}^{T} r_t(b_t) \right].$$

Following previous work (Feng et al., 2023; Li et al., 2025; Vijayan et al., 2025), we focus on the case of $\rho = 1$. Both the budget and RoS constraints can be handled via a primal-dual framework, but the budget constraint is a packing constraint, whereas the RoS constraint is non-packing. This distinction leads to substantive differences in how we control the dual regret.

**Assumption 2.1** (Unconfoundedness). The potential outcomes and the highest competing bid are conditionally independent given the context:

$$(v_{t,1}, v_{t,0}) \perp\!\!\!\perp m_t \mid x_t.$$

**Assumption 2.2** (Conditional Independence). Conditional on the sequence of contexts $\{x_t\}_{t=1}^{T}$, the sequence of random vectors $\{(v_{t,1}, v_{t,0}, m_t)\}_{t=1}^{T}$ is independent across time.

**Assumption 2.3** (Bounded Density). The cumulative distribution function (CDF) $F_t$ is $L$-Lipschitz continuous. That is, it admits a probability density function $f_t$ such that $\|f_t\|_\infty \leq L$.

**Assumption 2.4** (Noise Property). The noise $\xi_t$ is Gaussian with zero mean.

**Assumption 2.5** (Linear Growth Budget). The budget $B$ grows linearly with the time horizon $T$.

**Assumption 2.6** (Strong Slater Condition). There exists a fixed stationary policy $\pi_{\mathrm{slater}}$ and a constant margin $\delta > 0$ such that the expected constraint slack is strictly positive on average over the horizon:

$$\frac{1}{T} \sum_{t=1}^{T} \mathbb{E}[g_t(\pi_{\mathrm{slater}}(x_t))] \geq \delta, \quad (9)$$

where $g_t(b) = r_t(b) - c_t(b)$ is the constraint function when the target RoS is $\rho = 1$.

**Assumption 2.7** (Context Diversity and Propensity Overlap). The context distribution and the sequence of competitor distributions $\{F_t\}_{t=1}^{T}$ satisfy:

1. **Context Diversity:** The second moment matrix of the context is positive definite: $\mathbb{E}_{x \sim \mathcal{D}}[xx^\top] \succeq \kappa_x I_d$ with $\kappa_x > 0$.

2. **Sufficient Overlap (Uncertainty):** There exists a constant $c_{\mathrm{var}} > 0$ such that for any round $t$, the expected variance of the allocation indicator under the uniform bidding policy is bounded away from zero:

$$\mathbb{E}_{b \sim \pi_{\mathrm{uni}}}[F_t(b)(1 - F_t(b))] \geq c_{\mathrm{var}}. \quad (10)$$

We define the effective minimum eigenvalue constant as $\kappa := \kappa_x \cdot c_{\mathrm{var}}$.

We make a brief discussion of the above assumptions. Theorems 2.1 to 2.3 are adopted from Wen et al. (2025). Assumption 6.1 in Wen et al. (2025) is slightly more general than Assumption 2.4, but we adopt the Gaussian specification here for ease of exposition. Assumption 2.5 is common in online auction papers and follows from Balseiro & Gur (2019); Wang et al. (2023). For regret analysis under the

RoS constraint, an interaction term—the product of the Lagrange multiplier and the confidence width—can be unbounded. We assume a "Sufficient Overlap" condition to ensure that a burn-in phase can estimate the Slater constant to high precision. The "Sufficient Overlap" condition need only hold in expectation, which is considerably weaker than standard overlap assumptions in causal inference (Imbens & Rubin, 2015).

### 2.5. Main Results

We summarize our technical findings in the following theorems.

**Theorem 2.8.** *Suppose Theorems 2.1 to 2.4, there exists a bidding algorithm that achieves*

$$R_T^{Unc} = \tilde{O}(d\sqrt{T})$$

*in the unconstrained setting.*

**Theorem 2.9.** *Suppose Theorems 2.1 to 2.5, there exists a bidding algorithm that achieves*

$$R_T^{Bgt} = \tilde{O}(d\sqrt{T})$$

*in the budget constrained setting.*

**Theorem 2.10.** *Suppose Theorems 2.1 to 2.4, 2.6 and 2.7, there exists a bidding algorithm that achieves*

$$R_T^{RoS} = \tilde{O}(d\sqrt{T})$$

*in the RoS constrained setting.*

Since the $\Omega(\sqrt{dT})$ lower bound holds even for the unconstrained setting, our algorithms have minimax-optimal rates in terms of $T$.

## 3. Estimating the Competing Bid Distribution

By Equation (3), we know that the competing bid is linearly generated by a context vector $x_t$, and we first discuss how to estimate the competing bid distribution $F_t$ based on this structure.

Wen et al. (2025) show that under Theorems 2.1 to 2.4, the following lemma holds.

**Lemma 3.1** (Bernstein-Type Oracle Bound). *Under Theorems 2.1 to 2.4, and consider $\Sigma_t = \lambda I + \sum_{s=1}^{t-1} x_s x_s^\top$, suppose there exists a subset $S \in [t-1]$ such that $|S| \leq \frac{t-1}{2}$ and $\lambda I + \sum_{s \in S} x_s x_s^T \geq \frac{\Sigma_t}{9}$. Then, with probability at least $1 - T^{-1}$, there exists a sequence of error bounds $\{\epsilon_t\}_{t=1}^T$ such that for all $b \in [0, 1]$ and $t \in [T]$:*

$$|\hat{F}_t(b) - F_t(b)| \leq \epsilon_t \sqrt{F_t(b)(1 - F_t(b))} + \epsilon_t^2.$$

*where $\epsilon_t = O(\log T \sqrt{d/t} + \log T \|x_t\|_{\Sigma_t^{-1}})$ and $\Sigma_t = \lambda I + \sum_{s=1}^{t-1} x_s x_s^\top$.*

A limitation of Lemma 3.1 is its reliance on Lemma D.8, a technical result grounded in the work of Marcus et al. (2015). Lemma D.8 establishes that for any set of vectors, there exists a subset (of size at most half) whose aggregate covariance matrix spectrally lower-bounds a constant fraction of the total covariance. However, the proof in Marcus et al. (2015) is non-constructive, meaning it is unknown whether the subset selection required for Lemma 3.1 can be implemented efficiently. To address this, we propose Algorithm 1, a computationally efficient procedure based on the random flipping method in Lemma 3.2.

---

**Algorithm 1** Split-Sample CDF Estimation

---

1: **Input:** History of contexts $x_1, \ldots, x_{t-1}$, corresponding outcomes $m_1, \ldots, m_{t-1}$, and current context $x_t$.
2: **Output:** Estimated CDF $\hat{F}_t$.
3: **1. Data Splitting:**
4: Initialize training set $S_t \leftarrow \emptyset$ and residual set $S_t^c \leftarrow \emptyset$.
5: **for** $s = 1$ to $t - 1$ **do**
6:     Sample $z_s \sim \text{Bernoulli}(1/2)$.
7:     **if** $z_s = 1$ **then**
8:         $S_t \leftarrow S_t \cup \{s\}$ {Include in Training Set}
9:     **else**
10:         $S_t^c \leftarrow S_t^c \cup \{s\}$ {Include in Evaluation Set}
11:     **end if**
12: **end for**
13: **2. Parameter Estimation:**
14: Estimate parameter vector $\hat{\phi}_t$ using only the data in $S_t$ via OFUL.
15: **3. CDF Construction:**
16: Define the empirical CDF $\hat{F}_t(b)$ using the residual set $S_t^c$:

$$\hat{F}_t(b) \leftarrow \frac{1}{|S_t^c|} \sum_{s \in S_t^c} \mathbb{I}\left[m_s - \hat{\phi}_t^\top x_s + \hat{\phi}_t^\top x_t \leq b\right]$$

17: **Return** Function $\hat{F}_t$.

---

**Lemma 3.2** (Spectral Approximation via Random Splitting). *Let $x_1, \ldots, x_{t-1} \in \mathbb{R}^d$ be an arbitrary sequence of feature vectors with $\|x_s\|_2 \leq L$. Let $\Sigma_t = \lambda I + \sum_{s=1}^{t-1} x_s x_s^\top$.*

*Construct a training set $S_t \subseteq [t-1]$ by including each index $s$ independently with probability $1/2$. Define the local covariance matrix as:*

$$A_t = \frac{\lambda}{2}I + \sum_{s \in S_t} x_s x_s^\top. \tag{11}$$

*Then, for any $\delta > 0$, provided that the regularization parameter satisfies $\lambda \geq 16L^2 \log(d/\delta)$, with probability at least $1 - \delta$:*

$$A_t \succeq \frac{1}{4}\Sigma_t. \tag{12}$$

Combing Lemma 3.2 with Lemma 3.1 implies we can use Algorithm 1 to get an estimated CDF $\hat{F}_t$ with error bounded by $\epsilon_t = O(\log T \sqrt{d/t} + \log T \|x_t\|_{\Sigma_t^{-1}})$. We use roughly one half of the context vectors for learning the unknown parameter $\phi_\star$ and use the remaining context vectors to estimate the CDF.

## 4. The Primal-dual Framework and the Budget Constrained/Unconstrained Setting

---
**Algorithm 2** Unified OFUL-FPA Framework
---
1: **Input**: Horizon $T$, Mode $\in \{$Unc, Budget, RoS$\}$.
2: **Initialize**: Estimator stats $(A_0, u_0)$, Duals $(\mu_1, \lambda_1)$.
3: **for** $t = 1$ to $T$ **do**
4:     **1. Estimation**: Get $(\hat{F}_t, \epsilon_t)$ and $(\hat{\theta}_{t-1}, \beta_t)$.
5:     **2. Primal Planning (Switch Logic):**
6:     **if** Mode == Unc **then**
7:         $\gamma_t \leftarrow 1$
8:         $b_t \leftarrow$ PLANSTANDARD$(\ldots, \gamma_t)$
9:     **else if** Mode == Budget **then**
10:        $\gamma_t \leftarrow Z \cdot \mu_t$
11:       $b_t \leftarrow$ PLANSTANDARD$(\ldots, \gamma_t)$
12:     **else if** Mode == RoS **then**
13:        $\gamma_t \leftarrow \frac{\lambda_t}{1+\lambda_t}$
14:       $b_t \leftarrow$ PLANROS$(\ldots, \gamma_t)$
15:     **end if**
16:     **3. Execution and Dual Update:**
17:     Execute $b_t$. Update stats $A_t, u_t$.
18:     **if** Mode == Budget **then**
19:        $\mu_{t+1} \leftarrow \Pi_{[0,1]} \mu_t \exp(\eta(c_t - B/T))$
20:     **else if** Mode == RoS **then**
21:        $\lambda_{t+1} \leftarrow \Pi_{[0,\Lambda]} \lambda_t \exp(-\eta(\tilde{r}_t - \tilde{c}_t))$
22:     **end if**
23: **end for**
---

Next, we present our primal-dual framework (Algorithm 2) for handling the unconstrained and the budget/RoS constraint settings. The linear treatment effect can be handled using either the master framework (Chu et al., 2011) or OFUL (Abbasi-Yadkori et al., 2011). We adopt OFUL in our primal-dual framework for two reasons: (i) it is not clear whether linear contextual bandits with knapsack constraints can be accommodated within the master framework; (ii) the master framework is elimination-based, whereas under a budget constraint, an arm that is initially suboptimal may become optimal as the remaining budget decreases. Across different problem settings, we use different shadow prices $\gamma_t$, and for the RoS constrained setting, we need to revise the primal algorithm due to the non-convex nature of the constraint.

To keep our algorithms modular, we use several helper functions. For example, in Line 4, Algorithm 2 invokes Algo-

rithms 1 and 6 to estimate the required parameters. Then through Lines 6–14, Algorithm 2 selects the shadow price $\gamma_t$ for each setting by invoking the corresponding primal minimizer. In Line 17, Algorithm 2 invokes Algorithm 7 to estimate the variance, apply a truncated IPW estimator for the propensity score, and update parameters based on the resulting precision. Also, to ensure the Largrangian multiplier is bounded, in Lines 18-22, we project the Largrangian multiplier onto a closed interval. This operation is standard in online auctions with budget constraints (Wang et al., 2023) but is less common under RoS constraints (Feng et al., 2023; Li et al., 2025). The reason is that the dual regret is always bounded even if the Largrangian multiplier is very large, but the primal regret can be unbounded due to the estimation error. We therefore use an initial burn-in phase to estimate the Slater constant and, in turn, an upper bound on the optimal Lagrangian multiplier.

---
**Algorithm 3** Standard Primal Oracle (For Unconstrained & Budget)
---
**Function** PLANSTANDARD$(x_t, \hat{\theta}_{t-1}, A_{t-1}, \hat{F}_t, \epsilon_t, \beta_t, \gamma_t)$

1: **Initialize:** Max value $J^* \leftarrow -\infty$, Best bid $b^* \leftarrow 0$.
2: **for all** $b \in \mathcal{B}_K$ **do**
3:     Gap $\leftarrow \hat{\theta}_{t-1}^\top x_t - b$
4:     Rad $\leftarrow \beta_t \|x_t\|_{A_{t-1}^{-1}}$
5:     $u_{t,0} \leftarrow \hat{F}_t(b)(\text{Gap} + \text{Rad}) + 2\epsilon_t$
6:     $u_{t,1} \leftarrow \hat{F}_t(b)(\text{Gap}) - \hat{\theta}_{t-1}^\top x_t + (1 - \hat{F}_t(b))\text{Rad} + \hat{\theta}_{t-1}^\top x_t + 2\epsilon_t$
7:     $\tilde{r}_t \leftarrow \min(u_{t,0}, u_{t,1})$
8:     $\tilde{c}_t \leftarrow b \cdot \hat{F}_t(b)$
9:     $\tilde{c}_t^{LCB} \leftarrow \max(0, \tilde{c}_t - b\epsilon_t)$
10:    $J \leftarrow \tilde{r}_t - \gamma_t \cdot \tilde{c}_t^{LCB}$
11:    **if** $J > J^*$ **then**
12:       $J^* \leftarrow J, \; b^* \leftarrow b$
13:    **end if**
14: **end for**
15: **Return**
---

We now turn to Algorithm 3, the primal routine for the unconstrained and budget-constrained settings. The core idea follows from a clever "better of two UCBs" idea introduced in Wen et al. (2025). The standard UCB estimator $u_{t,0}$ has its error scale driven by $\hat{F}_t$, which becomes loose as $\hat{F}_t \to 1$ (the "loss of optimism" problem; see Line 5 of Algorithm 7 for intuition). The alternative estimator $u_{t,1}$ exploits the identity $r(b) = \theta_\star^T x - (1 - F)(\theta_\star^T x) - F(b)$ to shift the error sensitivity to $(1 - \hat{F}_t)$, making it well-behaved when $\hat{F}_t \to 1$. Taking the minimum $\min(u_{t,0}, u_{t,1})$ ensures that the effective confidence width scales with the variance proxy $\min(\hat{F}_t, 1 - \hat{F}_t) \approx \hat{F}_t(1 - \hat{F}_t)$, which is essential for a $\tilde{O}(\sqrt{T})$ regret bound.

We provide the following regret guarantee for online first-price auctions with linear treatment effects under the unconstrained/budget-constrained settings. The proof of Theorem 4.1 can be found in Appendix B.1

**Theorem 4.1** (Regret Bound for Primal-Dual OFUL-FPA). *Under Assumptions 2.1, 2.2, 2.3, and 2.5, let $Z = T/B$ be the budget normalization factor. With probability at least $1 - 2\delta$, the cumulative regret of Algorithm 2 under the Budget mode is bounded by:*

$$R_T^{Bgt} \leq \tilde{O}\left((d + Z)\sqrt{T}\right), \tag{13}$$

*where $\tilde{O}(\cdot)$ hides logarithmic factors in $T$, $d$, $1/\delta$, and algorithm constants.*

*Proof.* We define the total regret as $R_T^{\text{Bgt}} = OPT - \sum_{t=1}^{\tau-1} r_t(b_t)$. According to the Primal-Dual framework, we decompose the Regret into three parts:

$$R_T^{\text{Bgt}} = \underbrace{\left[OPT - \sum_{t=1}^{\tau-1} \tilde{r}_t(b_t) + Z\sum_{t=1}^{\tau-1} \mu_t\left(\tilde{c}_t(b_t) - \frac{B}{T}\right)\right]}_{\textbf{Term A: Primal Optimism Gap (Stop-Time Loss)}}$$

$$+ \underbrace{Z\sum_{t=1}^{\tau-1} \mu_t\left(\frac{B}{T} - c_t(b_t)\right)}_{\textbf{Term B: Dual Regret}}$$

$$+ \underbrace{\sum_{t=1}^{\tau-1} (\tilde{r}_t(b_t) - r_t(b_t)) + Z\sum_{t=1}^{\tau-1} \mu_t(c_t(b_t) - \tilde{c}_t(b_t))}_{\textbf{Term C: LTE Estimation Error}}$$

$$\tag{14}$$

The core challenge lies in the interaction between Term A and Term B at the random stopping time $\tau$. Term A leverages the optimistic Lagrangian to upper-bound the optimal policy (OPT) over the full horizon $T$, effectively "counting" rewards for rounds $t \geq \tau$ that the algorithm never actually executes. However, this overestimation is corrected by Term B: if the budget is exhausted early ($\tau < T$), the algorithm's spending rate necessarily exceeded the target rate $B/T$. This violation causes Term B (which tracks dual drift) to accumulate a large negative value proportional to the number of unplayed rounds $(T - \tau)$, effectively "canceling out" the unrealized rewards from Term A and ensuring the total regret remains sub-linear. Term C is bounded using standard martingale concentration inequalities and the weighted OFUL analysis to control the statistical errors from learning the unknown valuation parameters and bid distributions. □

Let $B \to \infty$, by Theorem 4.1, we automatically get the regret guarantee for the unconstrained case.

**Corollary 4.2.** *Under Assumptions 2.1, 2.2, 2.3, with probability at least $1 - 2\delta$, the cumulative regret of Algorithm 2 under the Unconstrained mode is bounded by:*

$$R_T^{Unc} \leq \tilde{O}\left(d\sqrt{T}\right), \tag{15}$$

*where $\tilde{O}(\cdot)$ hides logarithmic factors in $T$, $d$, $1/\delta$, and algorithm constants.*

# 5. Bidding under the RoS Constraint and the Adaptive Burn-in Procedure

Now we turn to the case of the RoS constraint. In a primal-dual framework, focusing on a deterministic strategy is often preferable. However, Li et al. (2025) find that restricting attention to deterministic strategies can be sub-optimal and propose a convexification procedure to handle the RoS constraint. Informally, Li et al. (2025) show that there exists a surrogate competing bid distribution whose allocation-payment curve corresponds to the convexification of the original competing bid distribution in the (payment, allocation) space. For this surrogate competing bid distribution, it suffices to consider a deterministic bidding strategy.

We present our primal algorithm for the RoS setting in Algorithm 4. Our algorithm differs from Li et al. (2025) in two key respects. Li et al. (2025) show the convexification argument works if the true competing bid distribution is known; accordingly, they split the time horizon into phases and apply a distribution estimator to estimate the competing bid distribution at the beginning of each phase. In our setting, we can estimate the CDF of the competing bid distribution via Algorithm 1 in each round; we therefore use a round-by-round adaptive estimation procedure in Algorithm 4. The second difference is that the value is hidden in our setting; accordingly, we still apply the "better of two UCBs" technique (Wen et al., 2025). Theorem 5.1 shows the regret bound of Algorithm 2 with RoS mode, and its proof can be found in Appendix C.1.

**Theorem 5.1** (Expected Regret and Violation Bounds with Fixed Projection). *Suppose Theorems 2.1 to 2.3, 2.6 and 2.7 hold. Let the algorithm run with a dual projection bound $\Lambda \geq 1$, step size $\eta = 1/\sqrt{T}$, and discretization grid size $K = \lceil\sqrt{T}\rceil$.*

*Then, the algorithm achieves the following expected performance bounds:*

***1. Expected Cumulative Regret:*** *The expected regret against any fixed benchmark policy $\pi^*$ compatible with the dual bound $\Lambda$ is bounded by:*

$$\mathbb{E}[R_T^{RoS}] \leq \tilde{O}\left(\Lambda \cdot d\sqrt{T}\right). \tag{16}$$

***2. Expected Cumulative RoS Violation:*** *The expected net*

*violation of the Return on Spend constraint is bounded by:*

$$\mathbb{E}[V_T^{RoS}] = \mathbb{E}\left[\sum_{t=1}^{T} -g_t(b_t)\right] \leq \tilde{O}\left(\sqrt{dT}\right). \qquad (17)$$

---

**Algorithm 4** Convexified Primal Oracle (For RoS Constraint)

---

1: **Function** PLANROS$(x_t, \hat{\theta}_{t-1}, A_{t-1}, \hat{F}_t, \epsilon_t, \beta_t, \gamma_t)$
2: **1. Deng's Transformation (Convexification):**
3: Construct Optimistic CDF: $F^\dagger(b) = \min(1, \hat{F}_t(b) + \epsilon_t)$.
4: Construct $F^{\dagger,\text{conv}}(b)$ based on the convexification of $F^\dagger(b)$ in the (Payment, Allocation) space.
5: **2. Grid Search with Dual UCBs:**
6: $J^* \leftarrow -\infty, b^* \leftarrow 0$
7: **for all** $b \in \mathcal{B}_K$ **do**
8:    Gap $\leftarrow \hat{\theta}_{t-1}^\top x_t - b$
9:    Rad $\leftarrow \beta_t \|x_t\|_{A_{t-1}^{-1}}$
10:   $u_{t,0} \leftarrow F^{\dagger,\text{conv}}(b)(\text{Gap} + \text{Rad}) + 2\epsilon_t$
11:   $u_{t,1} \leftarrow -(1-F^{\dagger,\text{conv}}(b))\hat{\theta}_{t-1}^\top x_t - bF^{\dagger,\text{conv}}(b) + (1-F^{\dagger,\text{conv}}(b))\text{Rad} + 2\epsilon_t$
12:   $\tilde{r}_t \leftarrow \min(u_{t,0}, u_{t,1})$
13:   $\tilde{c}_t \leftarrow b \cdot F^{\dagger,\text{conv}}(b)$
14:   $J \leftarrow \tilde{r}_t - \gamma_t \cdot \tilde{c}_t^{LCB}$
15:   **if** $J > J^*$ **then**
16:     $J^* \leftarrow J, \; b^* \leftarrow b$
17:   **end if**
18: **end for**
19: **Return** $b^*$

---

Theorem 5.1 assumes that we know a constant $\Lambda$ such that, with high probability, the optimal Largrangian multiplier lies in $[0, \Lambda]$; this requirement is impractical. We next show how to use a burn-in phase to remove this requirement. Before presenting the burn-in procedure, we state the following lemma, which shows that the reciprocal of the Slater constant provides an upper bound on the optimal Largrangian multiplier.

**Lemma 5.2** (Upper Bound on Optimal Dual Multiplier). *Under Assumption 2.6, let $\lambda^\star$ be the optimal dual multiplier for the hindsight optimization problem over $T$ rounds. Then:*

$$\lambda^\star \leq \frac{1}{\delta}. \qquad (18)$$

We adopt a uniform exploration strategy in the first $T_0$ rounds, which yields Lemma 5.3. The proof begins by establishing a lower bound on the estimated covariance matrix. Using Weyl's inequality, it shows that the spectral perturbation induced by approximating the true weights is dominated by the linear growth of the ideal covariance matrix. Once

the matrix is shown to be well-conditioned, we bound the parameter estimation error using standard self-normalized martingale concentration inequalities. We then convert the resulting matrix-norm bound to a Euclidean norm bound via the eigenvalue lower bound.

**Lemma 5.3** (Concentration via Adaptive Martingales with Estimated Weights). *Let $T_0 = \sqrt{T}$. Under Assumption 2.7, even if $\{F_t\}$ is an adaptive adversarial sequence and weights are estimated using $\hat{F}_t$, with probability at least $1 - 4/T$:*

1. ***Exploration Guarantee:** The estimated weighted covariance matrix $\hat{A}_{T_0} = \sum_{t=1}^{T_0} \hat{F}_t(b_t)(1-\hat{F}_t(b_t))x_t x_t^\top$ satisfies:*

$$\lambda_{\min}(\hat{A}_{T_0}) \geq \frac{\kappa T_0}{8}.$$

2. ***Estimation Error:** The parameter estimation error satisfies:*

$$\|\hat{\theta}_{T_0} - \theta_\star\|_2 \leq \tilde{O}(T^{-1/4}).$$

Lemma 5.3 essentially implies that as long as the Slater constant satisfies $\delta = \Omega\left(T^{-\frac{1}{4}}\right)$, our strategy can learn it efficiently, which aligns with Castiglioni et al. (2022b). With the help of Lemma 5.3, we have the following lemma, which guarantees the regret bound of Theorem 5.1 to be $\tilde{O}(d\sqrt{T})$. The proof of Lemma 5.3 and Lemma 5.4 can be found in Appendix C.3.

**Lemma 5.4** (Validity of Projection Interval). *Suppose the high-probability event in Lemma 5.3 holds (i.e., $\|\hat{\theta}_{T_0} - \theta_\star\|_2 \leq \tilde{O}(T^{-1/4})$). For a sufficiently large horizon $T$ (specifically $T \geq poly(1/\delta_{true})$), the projection bound $\Lambda = 2/\hat{\delta}$ constructed from the estimated parameter $\hat{\theta}_{T_0}$ satisfies:*

$$\lambda^\star \in [0, \Lambda], \qquad (19)$$

*where $\lambda^\star$ is the optimal dual multiplier for the hindsight problem.*

## Conclusion and Perspective

In this work, we conduct a systematic study of online first-price auctions with uplift values and the highest competing bid generated by a common context vector. Our main technical tool is a modularized primal-dual framework, and it applies to both the unconstrained setting and the budget/RoS constrained setting. Looking ahead, it is of interest to investigate tight regret guarantees in the partial feedback setting.

## Impact Statement

This paper presents work whose goal is to advance the field of Machine Learning. There are many potential societal consequences of our work, none which we feel must be specifically highlighted here.

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

## A. Proofs of Section.

### A.1. Proof of Lemma 3.2

Here we provide the proof of Lemma 3.2.

*Proof.* Let $G = \sum_{s=1}^{t-1} x_s x_s^\top$ be the full Gram matrix, and let $G_{S_t} = \sum_{s \in S_t} x_s x_s^\top$ be the subsampled Gram matrix. Define the random matrices $X_s = \xi_s x_s x_s^\top$, where $\xi_s \sim \text{Bernoulli}(1/2)$. Then $G_{S_t} = \sum_{s=1}^{t-1} X_s$. The expectation is:

$$M := \mathbb{E}[G_{S_t}] = \frac{1}{2} G.$$

We aim to prove that $A_t \succeq \frac{1}{4} \Sigma_t$. Substituting the definitions, this is equivalent to:

$$G_{S_t} + \frac{\lambda}{2} I \succeq \frac{1}{4}(G + \lambda I) \iff G_{S_t} \succeq \frac{1}{4} G - \frac{\lambda}{4} I.$$

Since $M = \frac{1}{2} G$, the condition is equivalent to showing:

$$G_{S_t} \succeq \frac{1}{2} M - \frac{\lambda}{4} I.$$

To prove this rigorously without approximation, we decompose the vector space $\mathbb{R}^d$ based on the eigenspaces of $M$. Let $v \in \mathbb{R}^d$ be any unit vector. It suffices to show $v^\top G_{S_t} v \geq \frac{1}{2} v^\top M v - \frac{\lambda}{4}$ for all $v$. We consider two cases for the magnitude of the signal $v^\top M v$:

**Case 1: Small Signal** ($v^\top M v \leq \lambda/2$). In this case, the RHS is $\frac{1}{2} v^\top M v - \frac{\lambda}{4} \leq \frac{\lambda}{4} - \frac{\lambda}{4} = 0$. Since $G_{S_t}$ is positive semi-definite, $v^\top G_{S_t} v \geq 0$. Thus, the inequality holds deterministically for all directions where the expected covariance is dominated by the regularization.

**Case 2: Large Signal** ($v^\top M v > \lambda/2$). Let $\mathcal{V}_{large}$ be the subspace spanned by eigenvectors of $M$ with eigenvalues strictly greater than $\lambda/2$. Let $P$ be the projection onto $\mathcal{V}_{large}$. We restrict our analysis to this subspace where the "signal" is strong enough for concentration. Within this subspace, the minimum eigenvalue of the expectation is bounded: $\lambda_{\min}(M|_{\mathcal{V}_{large}}) > \lambda/2$.

We apply the Matrix Chernoff Bound (Tropp, 2012) to the sequence $X_s$ restricted to this subspace. Let $R = L^2$ be the uniform bound ($\|X_s\|_2 \leq L^2$). For a relative error $\epsilon = 1/2$:

$$\mathbb{P}\left(\lambda_{\min}(PG_{S_t}P) \leq (1 - \epsilon)\lambda_{\min}(PMP)\right) \leq d \cdot \exp\left(-\frac{\epsilon^2 \lambda_{\min}(PMP)}{2R}\right).$$

Using the lower bound $\lambda_{\min}(PMP) > \lambda/2$ and substituting $\epsilon = 1/2$:

$$\text{Prob. of Failure} \leq d \cdot \exp\left(-\frac{(1/2)^2(\lambda/2)}{2L^2}\right) = d \cdot \exp\left(-\frac{\lambda}{16L^2}\right).$$

By our assumption $\lambda \geq 16L^2 \log(d/\delta)$, this probability is at most $\delta$. Thus, with probability $1 - \delta$, for all vectors in this subspace:

$$G_{S_t} \succeq \frac{1}{2} M \succeq \frac{1}{2} M - \frac{\lambda}{4} I.$$

**Conclusion:** Combining the two cases, the inequality holds for the entire space. Directions with small variance are protected by the regularization term $-\frac{\lambda}{4} I$ (making the requirement trivial), while directions with large variance are protected by the concentration of measure implied by the choice of $\lambda$.

Therefore, with probability at least $1 - \delta$:

$$A_t = G_{S_t} + \frac{\lambda}{2} I \succeq \left(\frac{1}{4} G - \frac{\lambda}{4} I\right) + \frac{\lambda}{2} I = \frac{1}{4} G + \frac{\lambda}{4} I = \frac{1}{4} \Sigma_t.$$

Inverting yields the result. $\square$

## A.2. Helper Algorithms

---

**Algorithm 5** Helper: Optimistic Reward Calculator (WLS Logic)

---

**Function** GETOPTIMISTICREWARD$(b, P, \text{Gap}, \text{Norm}, \beta_t)$

1: **Input:** Bid $b$, Probability $P$ (can be $\hat{F}$ or $F^\dagger$), Gap $(\hat{\theta}x - b)$, Norm $\|x\|_{A^{-1}}$, Radius $\beta_t$.
2: {Match width formula from Alg 1: Variance term + Higher-order term}
3: $V_{\text{proxy}} \leftarrow P(1 - P)$
4: $w_t \leftarrow c_3 L \beta_t \left( V_{\text{proxy}} \cdot \text{Norm} + \beta_t \cdot \text{Norm}^2 \right)$
5: $\tilde{r}_{\text{param}} \leftarrow P \cdot \text{Gap} + w_t$
6: **Return** $\tilde{r}_{\text{param}}$

---

**Algorithm 6** Helper: GetEstimates

---

**Function** GETESTIMATES$(x_t, \text{History}_{t-1}, A_{t-1}, u_{t-1}, \lambda, \delta)$

1: **Input:** Context $x_t$, Historical data, WLS stats, Hyperparams.
2: $\hat{\theta}_{t-1} \leftarrow A_{t-1}^{-1} u_{t-1}$
3: $\beta_t \leftarrow R\sqrt{d \log(1 + t/\lambda) + 2\log(1/\delta)} + \sqrt{\lambda}S$
4: **Return** $(\hat{\theta}_{t-1}, \beta_t)$

---

**Algorithm 7** Helper: UpdateStats

---

**Function** UPDATESTATS$(A_{t-1}, u_{t-1}, x_t, b_t, m_t, v_t, \hat{F}_t, \epsilon_t)$

1: **Input:** Old stats, context, bid $b_t$, competitor $m_t$, outcome $v_t \in \{v_{t,1}, v_{t,0}\}$.
2: **1. Calculate Variance Proxy (Weight):**
3: $\sigma_t^{-2} \leftarrow \hat{F}_t(b_t)(1 - \hat{F}_t(b_t))$
4: **2. Construct Transformed Outcome (for WLS):**
5: $\tilde{y}_t \leftarrow \frac{\mathbb{I}[b_t \geq m_t]}{\max\{\epsilon_t^2, \hat{F}_t(b_t)\}} v_{t,1} - \frac{\mathbb{I}[b_t < m_t]}{\max\{\epsilon_t^2, 1 - \hat{F}_t(b_t)\}} v_{t,0}$
6: **3. Recursive Update:**
7: $A_t \leftarrow A_{t-1} + \sigma_t^{-2} x_t x_t^\top$
8: $u_t \leftarrow u_{t-1} + \sigma_t^{-2} x_t \tilde{y}_t$
9: **Return** $(A_t, u_t)$

---

**Algorithm 8** Shared Helper: Variance-Safe Truncation

---

**Function** TRUNCATION$(b_{\text{raw}}, \hat{F}_t, \epsilon_t, \beta_t, T, d)$

1: $z \leftarrow \min\left\{ \beta_t \sqrt{\frac{d}{T}} + 4\epsilon_t, \frac{1}{2} \right\}$
2: $b_{\text{lower}} \leftarrow \hat{F}_t^{-1}(z)$
3: $b_{\text{upper}} \leftarrow \hat{F}_t^{-1}(1 - z)$
4: $b_{\text{safe}} \leftarrow \min\{\max\{b_{\text{raw}}, b_{\text{lower}}\}, b_{\text{upper}}\}$
5: **Return** $b_{\text{safe}}$

---

# B. Proof Details of Section 4

## B.1. Proof of Theorem 4.1

*Proof.* Recall the definition of Term A:

$$\text{Term A} = OPT - \sum_{t=1}^{\tau-1} \tilde{r}_t(b_t) + Z \sum_{t=1}^{\tau-1} \mu_t \left( \tilde{c}_t(b_t) - \frac{B}{T} \right). \tag{20}$$

Let $\mathcal{L}_t(b) = \tilde{r}_t(b) - Z\mu_t\tilde{c}_t(b)$ be the algorithm's optimistic Lagrangian. We can rewrite Term A as:

$$\text{Term A} = OPT - \sum_{t=1}^{\tau-1} \mathcal{L}_t(b_t) - Z\sum_{t=1}^{\tau-1} \mu_t \frac{B}{T}. \tag{21}$$

To upper bound Term A, we lower bound the cumulative Lagrangian $\sum \mathcal{L}_t(b_t)$. Since the algorithm selects $b_t = \arg\max_{b \in \mathcal{B}_K} \mathcal{L}_t(b)$, we have $\mathcal{L}_t(b_t) \geq \mathcal{L}_t(b_{\text{grid}})$, where $b_{\text{grid}} \in \mathcal{B}_K$ is the closest discretization point to the optimal continuous bid $b^*$.

Applying Lemma D.6 with weights $\alpha_t = 1 + \lambda_t$ and $\beta_t = \lambda_t$ where $\lambda_t := Z\mu_t$, the discretization gap for the Lagrangian is bounded by:

$$\mathcal{L}_t(b_t) \geq \mathcal{L}_t(b_{\text{grid}}) \geq \mathcal{L}_t(b^*) - \frac{L+1}{2K}(1 + 2\lambda_t). \tag{22}$$

The generalized Lagrangian is defined as $J_t(b) = \alpha_t \mathbb{E}[r_t(b)] - \beta_t \mathbb{E}[c_t(b)]$.

By the property of the "better of two UCBs" (Wen et al., 2025), the optimism gap ($\mathcal{L}_t \geq J_t$) is non-negative with high probability and is conservatively ignored here (absorbed into Term C). Summing over $t = 1 \ldots \tau - 1$:

$$\sum_{t=1}^{\tau-1} \mathcal{L}_t(b_t) \geq \sum_{t=1}^{\tau-1} (\mathbb{E}[r_t(b^*)] - Z\mu_t\mathbb{E}[c_t(b^*)]) - \frac{L+1}{2K}\sum_{t=1}^{\tau-1}(1 + Z\mu_t). \tag{23}$$

Plugging this lower bound back into the expression for Term A:

$$\text{Term A} \leq OPT - \left[\sum_{t=1}^{\tau-1} \mathbb{E}[r_t(b^*)] - Z\sum_{t=1}^{\tau-1} \mu_t\mathbb{E}[c_t(b^*)]\right] - Z\sum_{t=1}^{\tau-1} \mu_t \frac{B}{T} + \frac{L+1}{2K}\sum_{t=1}^{\tau-1}(1 + Z\mu_t)$$

$$= \underbrace{\left(OPT - \sum_{t=1}^{\tau-1} \mathbb{E}[r_t(b^*)]\right)}_{\text{Term D: Stopping Time Loss}} + \underbrace{Z\sum_{t=1}^{\tau-1} \mu_t\left(\mathbb{E}[c_t(b^*)] - \frac{B}{T}\right)}_{\text{Feasibility Term}} + \underbrace{\frac{L+1}{2K}\sum_{t=1}^{\tau-1}(1 + Z\mu_t)}_{\text{Discretization Error}}. \tag{24}$$

We now analyze the components:

- **Feasibility Term:** Since $\pi^*$ is feasible ($\mathbb{E}[c_t] \leq B/T$) and the dual variables are non-negative ($\mu_t \geq 0$), the term is non-positive.

- **Discretization Error:** Using $\mu_t \leq 1$ and $\tau \leq T + 1$, the error is bounded by $\frac{(L+1)T}{2K}(1 + Z)$.

- **Term D (Stopping Time Loss):** This term represents the reward "left on the table" due to early stopping. Since $x_t$ are i.i.d. and the policy is static, rewards accumulate linearly: $\sum_{t=1}^{\tau-1} \mathbb{E}[r_t(b^*)] = \frac{\tau-1}{T}OPT$, where the equality holds in expectation over the randomness of contexts.

Thus, we explicitly define Term D and bound Term A as:

$$\text{Term A} \leq \underbrace{OPT\left(1 - \frac{\tau-1}{T}\right)}_{\text{Term D}} + \frac{(L+1)T}{2K}(1 + Z). \tag{25}$$

We now incorporate Term B, the Dual Regret, which acts as a counterbalance to Term D.

$$\text{Term B} = Z\sum_{t=1}^{\tau-1} \mu_t\left(\frac{B}{T} - c_t(b_t)\right). \tag{26}$$

By Lemma D.2 (Dual Regret Bound), if the algorithm stops early ($\tau \leq T$), the budget is essentially exhausted ($\sum c_t \approx B$). The dual variables penalize this overspending:

$$\text{Term B} \leq -ZB\left(1 - \frac{\tau - 1}{T}\right) + \tilde{O}(Z\sqrt{T}). \tag{27}$$

Combining the bounds for Term A (Eq. 25) and Term B:

$$\begin{aligned}
\text{Term A} + \text{Term B} &\leq \left[OPT\left(1 - \frac{\tau - 1}{T}\right) + \text{Disc. Error}\right] \\
&\quad + \left[-ZB\left(1 - \frac{\tau - 1}{T}\right) + \tilde{O}(Z\sqrt{T})\right] \\
&= \underbrace{(OPT - ZB)}_{\leq 0} \cdot \underbrace{\left(1 - \frac{\tau - 1}{T}\right)}_{\geq 0} + \text{Disc. Error} + \tilde{O}(Z\sqrt{T}).
\end{aligned}$$

Crucially, since $Z = T/B$, we have $ZB = T$. Furthermore, assuming normalized rewards $r_t \in [0, 1]$, we have $OPT \leq T$. Therefore:

$$OPT - ZB \leq T - T = 0.$$

This implies that the Primal loss from stopping early (Term D) is fully compensated by the Dual penalty. The leading terms vanish, leaving only:

$$\text{Term A} + \text{Term B} \leq \frac{(L+1)T}{2K}(1 + Z) + \tilde{O}(Z\sqrt{T}) = \tilde{O}\left((1 + Z)\sqrt{T}\right). \tag{28}$$

(Setting $K = \sqrt{T}$).

Term C is the total estimation error accumulated up to the stopping time $\tau$:

$$\text{Term C} = \underbrace{\sum_{t=1}^{\tau-1}(\tilde{r}_t(b_t) - r_t(b_t))}_{\text{Reward Estimation}} + Z\underbrace{\sum_{t=1}^{\tau-1}\mu_t(c_t(b_t) - \tilde{c}_t(b_t))}_{\text{Cost Estimation}} \tag{29}$$

We define the conditional expected reward $\bar{r}_t(b_t) = \mathbb{E}[r_t(b_t) \mid \mathcal{F}_{t-1}]$. We decompose the error into Martingale Noise and Estimation Bias:

$$\sum_{t=1}^{\tau-1}(\tilde{r}_t - r_t) = \underbrace{\sum_{t=1}^{\tau-1}(\bar{r}_t(b_t) - r_t(b_t))}_{\text{Part A: Martingale Noise}} + \underbrace{\sum_{t=1}^{\tau-1}(\tilde{r}_t(b_t) - \bar{r}_t(b_t))}_{\text{Part B: Estimation Bias}}. \tag{30}$$

**Analysis of Part A (Martingale Noise)** Let $M_t = \bar{r}_t(b_t) - r_t(b_t)$. Since $b_t$ is $\mathcal{F}_{t-1}$-measurable, $\mathbb{E}[M_t \mid \mathcal{F}_{t-1}] = 0$. Thus, $\{M_t\}$ is a Martingale Difference Sequence (MDS) with $|M_t| \leq 1$.

We seek to bound the sum $\sum_{t=1}^{\tau-1} M_t$. Since $\tau$ is a random stopping time bounded by $T$, we can rewrite this sum using the indicator function $\mathbb{I}[t < \tau]$, which is $\mathcal{F}_{t-1}$-predictable:

$$\sum_{t=1}^{\tau-1} M_t = \sum_{t=1}^{T} M_t \cdot \mathbb{I}[t < \tau]. \tag{31}$$

The sequence $M'_t = M_t \cdot \mathbb{I}[t < \tau]$ remains a Martingale Difference Sequence with respect to $\mathcal{F}_t$, bounded by 1. Applying the Azuma-Hoeffding inequality to the fixed horizon $T$, with probability at least $1 - \delta/2$:

$$\sum_{t=1}^{\tau-1} M_t = \sum_{t=1}^{T} M'_t \leq \sqrt{2T\log(2/\delta)} = \tilde{O}(\sqrt{T}). \tag{32}$$

Using the definition of the optimistic reward $\tilde{r}_t = \hat{r}_t + w_{r,t} + B_t$ and the concentration event $|\hat{r}_t - \bar{r}_t| \leq w_{r,t} + B_t$, we have:

$$\tilde{r}_t(b_t) - \bar{r}_t(b_t) \leq (\hat{r}_t + w_{r,t} + B_t) - (\hat{r}_t - w_{r,t} - B_t) = 2w_{r,t}(b_t) + 2B_t. \tag{33}$$

Summing over the active rounds:

$$\text{Part B} \leq 2\sum_{t=1}^{\tau-1} w_{r,t}(b_t) + 2\sum_{t=1}^{\tau-1} B_t. \tag{34}$$

We now rigorously show that $\sum_{t=1}^{\tau-1} w_{r,t}(b_t) \leq \tilde{O}(d\sqrt{T})$, utilizing the Truncation Lemma (D.3) and the Elliptical Potential Lemma (D.4).

Wen et al. (2025) show that for the "better of two UCBs", the width is bounded by

$$w_{r,t}(b_t) = c_3 L\beta_t \left( \hat{F}_t(1 - \hat{F}_t)\|x_t\|_{A_t^{-1}} + \beta_t\|x_t\|_{A_t^{-1}}^2 \right). \tag{35}$$

Our goal is to bound the cumulative sum $\sum_{t=1}^{\tau-1} w_{r,t}(b_t)$.

We invoke the general result Lemma D.7 with variance proxy $\hat{V}_t = \hat{F}_t(b_t)(1 - \hat{F}_t(b_t))$. Since we have done the truncation operation in Algorithm 8 using the threshold $z = \min\left\{ \beta_t\sqrt{\frac{d}{T}} + 4\epsilon_t, \frac{1}{2} \right\}$, by invoking Lemma D.5, this specific choice of $z$ guarantees that the inverse variance proxy satisfies the rigorous bound:

$$\sigma_t^2(\tilde{b}_t) \leq \frac{2}{\beta_t}\sqrt{\frac{T}{d}} + 5. \tag{36}$$

The precondition of Lemma D.7 is satisfied by setting the constants $K = 2/\sqrt{d}$ and $\sigma_{\max}^2 = 5$. Therefore, the total sum of widths is bounded by:

$$\sum_{t=1}^{\tau-1} w_{r,t}(b_t) \leq \tilde{O}(\beta_T\sqrt{dT}). \tag{37}$$

Since $\beta_T = \tilde{O}(\sqrt{d})$, we obtain the final bound $\tilde{O}(d\sqrt{T})$.

We also need to bound the cumulative cost estimation error. We define the true conditional expected cost given the history $\mathcal{F}_{t-1}$ as:

$$\bar{c}_t(b_t) \triangleq \mathbb{E}[c_t(b_t) \mid \mathcal{F}_{t-1}] = b_t \cdot \mathbb{I}(b_t \geq m_t \mid \mathcal{F}_{t-1}) = b_t F_t(b_t). \tag{38}$$

We decompose the instantaneous error $(c_t - \tilde{c}_t)$ into a stochastic noise term and a prediction bias term:

$$c_t(b_t) - \tilde{c}_t(b_t) = \underbrace{(c_t(b_t) - \bar{c}_t(b_t))}_{\text{Martingale Difference}} + \underbrace{(\bar{c}_t(b_t) - \tilde{c}_t(b_t))}_{\text{Prediction Bias}}. \tag{39}$$

Substituting this back into the sum, we obtain:

$$\text{Term}_{\text{Est, Cost}} = \underbrace{Z\sum_{t=1}^{\tau-1} \mu_t(c_t(b_t) - \bar{c}_t(b_t))}_{\text{Part A}} + \underbrace{Z\sum_{t=1}^{\tau-1} \mu_t(\bar{c}_t(b_t) - \tilde{c}_t(b_t))}_{\text{Part B}}. \tag{40}$$

**Analysis of Part A (Martingale Concentration)** Let $M_t = Z\mu_t(c_t(b_t) - \bar{c}_t(b_t))$. It is easy to verify that $\{M_t\}_{t=1}^{\tau-1}$ forms a Martingale Difference Sequence (MDS) with respect to the filtration $\mathcal{F}_t$. Applying the Azuma-Hoeffding inequality, with probability at least $1 - \delta/2$:

$$\sum_{t=1}^{\tau-1} M_t \leq \sqrt{2\sum_{t=1}^{\tau-1} Z^2 \log(2/\delta)} \leq Z\sqrt{2T\log(2/\delta)} = \tilde{O}(Z\sqrt{T}). \tag{41}$$

**Analysis of Part B (Prediction Bias)** We analyze the deterministic difference between the true expected cost and the optimistic lower bound.

Recall the definition of the optimistic cost:

$$\tilde{c}_t(b_t) = \max\left(0, b_t \hat{F}_t(b_t) - b_t \epsilon_t\right). \tag{42}$$

We seek to bound $\Delta_t = \bar{c}_t(b_t) - \tilde{c}_t(b_t)$. Since $\tilde{c}_t$ involves a maximum function, we consider two cases:

- **Case 1:** The term is non-negative, i.e., $b_t \hat{F}_t(b_t) \geq b_t \epsilon_t$. In this case, $\tilde{c}_t(b_t) = b_t \hat{F}_t(b_t) - b_t \epsilon_t$.

$$\Delta_t = b_t F_t(b_t) - \left(b_t \hat{F}_t(b_t) - b_t \epsilon_t\right)$$

$$= b_t \left(F_t(b_t) - \hat{F}_t(b_t)\right) + b_t \epsilon_t. \tag{43}$$

Using the Oracle Assumption $|\hat{F}_t(b) - F_t(b)| \leq \epsilon_t$, we have $F_t - \hat{F}_t \leq \epsilon_t$. Since $b_t \in [0, 1]$:

$$\Delta_t \leq b_t(\epsilon_t) + b_t \epsilon_t = 2b_t \epsilon_t \leq 2\epsilon_t. \tag{44}$$

- **Case 2:** The term is clipped to zero, i.e., $b_t \hat{F}_t(b_t) < b_t \epsilon_t$. In this case, $\tilde{c}_t(b_t) = 0$. This condition implies $\hat{F}_t(b_t) < \epsilon_t$ (assuming $b_t > 0$, otherwise the cost is trivially 0). We bound the true expected cost:

$$\Delta_t = \bar{c}_t(b_t) - 0 = b_t F_t(b_t). \tag{45}$$

Using the Oracle Assumption again ($F_t \leq \hat{F}_t + \epsilon_t$):

$$F_t(b_t) < \epsilon_t + \epsilon_t = 2\epsilon_t. \tag{46}$$

Thus, $\Delta_t = b_t F_t(b_t) \leq 1 \cdot (2\epsilon_t) = 2\epsilon_t$.

In both cases, the bias is uniformly bounded by $2\epsilon_t$. Summing over $T$ and applying the bound $\mu_t \leq 1$:

$$Z \sum_{t=1}^{\tau-1} \mu_t(\bar{c}_t - \tilde{c}_t) \leq Z \sum_{t=1}^{\tau-1} 1 \cdot (2\epsilon_t) = 2Z \sum_{t=1}^{\tau-1} \epsilon_t = 2Z\Delta_{\text{oracle}}. \tag{47}$$

Combining the bounds from Part A and Part B, the total Cost Estimation Error is bounded by:

$$Z \sum_{t=1}^{\tau-1} \mu_t(c_t - \tilde{c}_t) \leq \tilde{O}(Z\sqrt{T}) + 2Z \sum_{t=1}^{\tau-1} \epsilon_t. \tag{48}$$

From our previous analysis, we have the following upper bounds for each term up to the stopping time $\tau$:

**Term A (Primal Optimism) & Term B (Dual Regret):**
Combining the cost of early stopping with the dual gain from the projected update (which is valid under Assumption 2.5):

$$\text{Term A} + \text{Term B} \leq \tilde{O}(Z\sqrt{T}). \tag{49}$$

**Term C (Estimation Error):**
This term consists of the Reward Estimation Error and the Cost Estimation Error.

- *Reward Part:* The martingale noise is $\tilde{O}(\sqrt{T})$ and the bias (width) sum is $\tilde{O}(d\sqrt{T})$.

- *Cost Part:* Using the projection $\mu_t \leq 1$, the martingale noise is $\tilde{O}(Z\sqrt{T})$ and the bias is bounded by $2Z \sum_{t=1}^{\tau-1} \epsilon_t$.

Summing these components:

$$\text{Term C} \leq \underbrace{\tilde{O}(d\sqrt{T})}_{\text{Reward Width}} + \underbrace{\tilde{O}(Z\sqrt{T})}_{\text{Cost Martingale}} + \underbrace{2Z \sum_{t=1}^{\tau-1} \epsilon_t}_{\text{Oracle Bias}} = \tilde{O}((d+Z)\sqrt{T}). \tag{50}$$

by considering $\epsilon_t = \tilde{O}\left(\frac{1}{\sqrt{t}}\right)$.

**Step 2: Final Summation** The total regret is the sum of these components:

$$R_T^{\text{Bgt}} \leq (\text{Term A} + \text{Term B}) + \text{Term C}$$

$$\leq \tilde{O}(Z\sqrt{T}) + \tilde{O}((d+Z)\sqrt{T}) \tag{51}$$

$$= \tilde{O}(d\sqrt{T}) \tag{52}$$

□

## C. Proof Details of Section 5

### C.1. Proof of Theorem 5.1

*Proof.* To align with the double optimism strategy, we define three distinct Lagrangians for round $t$. Let $\theta_\star^\top x_t$ denote the true valuation.

- **1. True Lagrangian ($\mathcal{L}_t$):** The benchmark using true distributions and parameters.

$$\mathcal{L}_t(b) = (1 + \lambda_t)(\theta_\star^\top x_t)F_t^{\text{conv}}(b) - \lambda_t b F_t^{\text{conv}}(b). \tag{53}$$

- **2. Bridge Lagrangian ($\mathcal{L}_t^\dagger$):** Replaces the true distribution with the optimistic one ($F^\dagger$), isolating distribution error.

$$\mathcal{L}_t^\dagger(b) = (1 + \lambda_t)(\theta_\star^\top x_t)F_t^{\dagger,\text{conv}}(b) - \lambda_t b F_t^{\dagger,\text{conv}}(b). \tag{54}$$

- **3. Fully Optimistic Lagrangian ($\tilde{\mathcal{L}}_t$):** Further replaces the true valuation with the UCB estimate ($\tilde{r}_t$). This is the actual objective maximized by the algorithm.

$$\tilde{\mathcal{L}}_t(b) = (1 + \lambda_t)\underbrace{\left[(\hat{\theta}_t^\top x_t)F_t^{\dagger,\text{conv}}(b) + w_t(b)\right]}_{\tilde{r}_t(b)} - \lambda_t b F_t^{\dagger,\text{conv}}(b). \tag{55}$$

The proof is divided into two parts: Regret Analysis and Violation Analysis. We leverage the linearity of expectation to aggregate the bounds derived in the technical lemmas below.

**Part 1: Regret Bound Analysis.** We start with the full Lagrangian Regret Decomposition (in expectation):

$$\mathbb{E}[R(T)] = \mathbb{E}\left[\sum_{t=1}^T \left(\mathcal{L}_t(\pi^*, \lambda_t) - \mathcal{L}_t(\tilde{b}_t, \lambda_t)\right)\right] + \mathbb{E}\left[\sum_{t=1}^T \lambda_t g_t(b_t)\right] \underbrace{-\mathbb{E}\left[\sum_{t=1}^T \lambda_t g_t(\pi^*)\right]}_{\text{Term C: Benchmark Slack}}$$

$$\leq \underbrace{\mathbb{E}\left[\sum_{t=1}^T \left(\mathcal{L}_t(\pi^*, \lambda_t) - \mathcal{L}_t(\tilde{b}_t, \lambda_t)\right)\right]}_{\text{Term A: Primal Gap}} + \underbrace{\mathbb{E}\left[\sum_{t=1}^T \lambda_t g_t(b_t)\right]}_{\text{Term B: Dual Drift}}. \tag{56}$$

The inequality holds because the benchmark policy $\pi^*$ is feasible in expectation ($\mathbb{E}[g_t(\pi^*)] \geq 0$) and dual variables are non-negative ($\lambda_t \geq 0$), implying the slack term is non-positive ($-\mathbb{E}[\lambda_t g_t(\pi^*)] \leq 0$).

**Bounding Term A (Primal Gap):** We invoke Lemma C.1, which aggregates distribution estimation errors, parameter estimation errors (via Weighted OFUL), and discretization errors. Specifically, it establishes:

$$\text{Term A} \leq \tilde{O}\left((1 + \Lambda)d\sqrt{T}\right). \tag{57}$$

**Bounding Term B (Dual Drift):** We explicitly derive the bound for the dual drift term. Using the update rule $\lambda_{t+1} = \Pi_{[0,\Lambda]}(\lambda_t \exp(-\eta g_t))$, non-expansiveness of projection implies $\lambda_{t+1} \leq \lambda_t \exp(-\eta g_t)$. Applying the inequality $e^{-x} \leq 1 - x + x^2$ (valid for bounded $|x|$):

$$\lambda_{t+1} \leq \lambda_t(1 - \eta g_t + \eta^2 g_t^2). \tag{58}$$

Rearranging to isolate the drift term $\lambda_t g_t$:

$$\eta \lambda_t g_t \leq \lambda_t - \lambda_{t+1} + \eta^2 \lambda_t g_t^2. \tag{59}$$

Dividing by $\eta$ and summing over $t = 1$ to $T$:

$$\sum_{t=1}^{T} \lambda_t g_t \leq \frac{1}{\eta} \sum_{t=1}^{T} (\lambda_t - \lambda_{t+1}) + \eta \sum_{t=1}^{T} \lambda_t g_t^2$$

$$= \frac{\lambda_1 - \lambda_{T+1}}{\eta} + \eta \sum_{t=1}^{T} \lambda_t g_t^2. \tag{60}$$

Using the telescopic sum property $\lambda_1 - \lambda_{T+1} \leq \Lambda$, the bound $\lambda_t \leq \Lambda$, and $|g_t| \leq 1$:

$$\sum_{t=1}^{T} \lambda_t g_t \leq \frac{\Lambda}{\eta} + \eta T \Lambda. \tag{61}$$

Substituting $\eta = 1/\sqrt{T}$:

$$\text{Term B} = \sum_{t=1}^{T} \lambda_t g_t \leq \Lambda\sqrt{T} + \Lambda\sqrt{T} = 2\Lambda\sqrt{T}. \tag{62}$$

**Total Regret:** Summing the bounds for Term A and Term B:

$$\mathbb{E}[R(T)] \leq \tilde{O}\left((1+\Lambda)d\sqrt{T}\right) + 2\Lambda\sqrt{T} = \tilde{O}\left(\Lambda d\sqrt{T}\right). \tag{63}$$

**Part 2: Violation Bound Analysis.** The expected net violation is bounded by the sum of the "Planned Violation" and the "Execution Error":

$$\mathbb{E}[V(T)] \leq \mathbb{E}\left[\sum_{t=1}^{T} -g_t(b_t)\right] \leq \underbrace{\mathbb{E}\left[\sum_{t=1}^{T} -\tilde{g}_t(\tilde{b}_t)\right]}_{\text{Planned Violation}} + \underbrace{\mathbb{E}\left[\sum_{t=1}^{T} |g_t(b_t) - \tilde{g}_t(\tilde{b}_t)|\right]}_{\text{Cumulative Error}}. \tag{64}$$

**Bounding Planned Violation:** We invoke Lemma C.3, which leverages the exponential growth of the dual variable to suppress large violations:

$$\text{Planned Violation} \leq \sqrt{T} \ln T + 2. \tag{65}$$

**Bounding Cumulative Error:** We invoke Lemma C.4, which bounds the difference between planned and executed constraint values using confidence widths and truncation analysis:

$$\text{Cumulative Error} \leq \tilde{O}\left(\sqrt{dT}\right). \tag{66}$$

**Total Violation:** Summing the components:

$$\mathbb{E}[V(T)] \leq (\sqrt{T} \ln T + 2) + \tilde{O}(\sqrt{dT}) = \tilde{O}\left(\sqrt{dT}\right). \tag{67}$$

$\square$

**Lemma C.1** (Bound on Expected Primal Gap (Term A)). *Assume the Oracle Error satisfies $\epsilon_t \leq O(t^{-1/2})$ and the discretization grid size is $K = \sqrt{T}$. The expected Primal Gap is decomposed into Distribution Error (A1+A2), Parameter Estimation Error (A3), and Discretization Error (A4). It is bounded by:*

$$\mathbb{E}[\text{Term A}] \leq \tilde{O}\left((1+\Lambda)\sqrt{T}\right) + \tilde{O}\left((1+\Lambda)d\sqrt{T}\right) + O(\Lambda\sqrt{T}) = \tilde{O}\left((1+\Lambda)d\sqrt{T}\right). \tag{68}$$

*Proof.* Recall that Term A is upper bounded by $\sum_{t=1}^{T}(\mathcal{L}_t(b_t^*) - \mathcal{L}_t(\tilde{b}_t))$, where $b_t^*$ is the optimal bid under the true environment and $\tilde{b}_t$ is the bid chosen by the algorithm. We insert the Bridge Lagrangian $\mathcal{L}_t^\dagger$ and the Fully Optimistic Lagrangian $\tilde{\mathcal{L}}_t$ to decompose the regret:

Let $\tilde{b}_t \in \mathcal{B}_K$ be the bid chosen by the algorithm (the maximizer over the discrete grid). We insert the Bridge Lagrangian $\mathcal{L}_t^\dagger$ and the Fully Optimistic Lagrangian $\tilde{\mathcal{L}}_t$ to decompose the regret, while explicitly accounting for the discretization gap.

$$
\begin{aligned}
\text{Term A} &= \sum_{t=1}^{T} \left( \mathcal{L}_t(b_t^*) - \mathcal{L}_t(\tilde{b}_t) \right) \\
&= \sum_{t=1}^{T} \Bigg[ \underbrace{\left( \mathcal{L}_t(b_t^*) - \mathcal{L}_t^\dagger(b_t^*) \right)}_{\textbf{(A1) Dist. Error at } b_t^*} + \underbrace{\left( \mathcal{L}_t^\dagger(b_t^*) - \tilde{\mathcal{L}}_t(b_t^*) \right)}_{\leq 0 \text{ (Param Optimism)}} \\
&\quad + \underbrace{\left( \tilde{\mathcal{L}}_t(b_t^*) - \tilde{\mathcal{L}}_t(\tilde{b}_t) \right)}_{\textbf{(A4) Discretization Error}} \\
&\quad + \underbrace{\left( \tilde{\mathcal{L}}_t(\tilde{b}_t) - \mathcal{L}_t^\dagger(\tilde{b}_t) \right)}_{\textbf{(A3) Param. Est. Error}} + \underbrace{\left( \mathcal{L}_t^\dagger(\tilde{b}_t) - \mathcal{L}_t(\tilde{b}_t) \right)}_{\textbf{(A2) Dist. Error at } \tilde{b}_t} \Bigg].
\end{aligned}
\tag{69}
$$

Terms (A1) and (A2) are handled by Lemma C.2. Term (A3) represents the cumulative parameter estimation error. Using the definition of the optimistic Lagrangian $\tilde{\mathcal{L}}_t$ and the Bridge Lagrangian $\mathcal{L}_t^\dagger$, this error is bounded by the cumulative confidence width weighted by the dual variable:

$$
(A3) = \sum_{t=1}^{T} \left( \tilde{\mathcal{L}}_t(\tilde{b}_t) - \mathcal{L}_t^\dagger(\tilde{b}_t) \right) \leq \sum_{t=1}^{T} (1 + \lambda_t) w_t(\tilde{b}_t) \leq (1 + \Lambda) \sum_{t=1}^{T} w_t(\tilde{b}_t).
\tag{70}
$$

To bound the sum of widths $\sum_{t=1}^{T} w_t(\tilde{b}_t)$, we invoke Lemma D.7. We verify that the conditions of the Lemma are satisfied by our algorithm:

Due to the "better of two UCBs" estimator (Wen et al., 2025), the width is bounded by

$$
w_t(b_t) = c_3 L \beta_t \left( \hat{F}_t(1 - \hat{F}_t) \|x_t\|_{A_t^{-1}} + \beta_t \|x_t\|_{A_t^{-1}}^2 \right).
\tag{71}
$$

And we can apply Lemma Lemma D.7 directly to get

$$
\sum_{t=1}^{T} w_t(\tilde{b}_t) \leq \tilde{O}\left( \beta_T \sqrt{dT} \right).
\tag{72}
$$

Substituting $\beta_T = \tilde{O}(\sqrt{d})$ (for linear kernels), we obtain:

$$
(A3) \leq (1 + \Lambda) \cdot \tilde{O}(d\sqrt{T}) = \tilde{O}((1 + \Lambda)d\sqrt{T}).
\tag{73}
$$

The term (A4) represents the loss due to optimizing over a finite grid $\mathcal{B}_K$ instead of the continuous interval $[0, 1]$. Since $\tilde{b}_t = \operatorname{argmax}_{b \in \mathcal{B}_K} \tilde{\mathcal{L}}_t(b)$, this term is exactly the difference between the continuous maximum and the discrete maximum of the function $\tilde{\mathcal{L}}_t(\cdot)$.

Applying Lemma D.6 with the Lagrangian weights $\alpha_t = 1 + \lambda_t$ and $\beta_t = \lambda_t$:

$$
\begin{aligned}
(A4) &= \sum_{t=1}^{T} \left( \max_{b \in [0,1]} \tilde{\mathcal{L}}_t(b) - \max_{b \in \mathcal{B}_K} \tilde{\mathcal{L}}_t(b) \right) \\
&\leq \sum_{t=1}^{T} \frac{L+1}{2K} ((1 + \lambda_t) + \lambda_t) \\
&\leq \sum_{t=1}^{T} \frac{L+1}{2K} (1 + \Lambda(1+)) \\
&= O\left( \frac{\Lambda T}{K} \right).
\end{aligned}
\tag{74}
$$

By choosing the grid size $K = \lceil \sqrt{T} \rceil$, the total discretization error is bounded by $O(\sqrt{T})$, matching the order of the other regret terms. $\square$

**Lemma C.2** (Distribution Estimation Error Bound). *Under the Oracle Assumption, the cumulative regret contributions from distribution estimation errors, specifically Term (A1) and Term (A2), are bounded as follows:*

$$
(A1) + (A2) \leq \tilde{O}\left( (1 + \Lambda) \sqrt{T} \right).
$$

*Proof.* The proof proceeds in three steps: bounding the error of the convex envelopes, establishing the Lipschitz property of the Lagrangian with respect to the distribution, and summing the errors over the horizon.

**Step 1: Non-expansiveness of the Convex Envelope.** Let $F_t$ and $F_t^{\dagger}$ be the true and optimistic distributions, respectively. From $F^{\dagger}(b) = \min(1, \hat{F}_t(b) + \epsilon_t)$ and Lemma 3.1, we have the uniform bound:

$$
\sup_{b \in [0,1]} |F_t^{\dagger}(b) - F_t(b)| \leq C\epsilon_t.
$$

A known property of the convex (or concave) envelope operator is that it is non-expansive in the $L_{\infty}$ norm. Specifically, if $f$ and $g$ are bounded functions, then $\|\text{conv}(f) - \text{conv}(g)\|_{\infty} \leq \|f - g\|_{\infty}$. Applying this to our setting:

$$
\sup_{b \in [0,1]} \left| F_t^{\dagger,\text{conv}}(b) - F_t^{\text{conv}}(b) \right| \leq \sup_{b \in [0,1]} \left| F_t^{\dagger}(b) - F_t(b) \right| \leq C\epsilon_t.
\tag{75}
$$

**Step 2: Lagrangian Difference Bound.** We analyze the difference between the True Convexified Lagrangian $\mathcal{L}_t$ and the Bridge Lagrangian $\mathcal{L}_t^{\dagger}$ for an arbitrary bid $b$. Recall the definitions from the Analysis of Term A :

$$
\begin{aligned}
\mathcal{L}_t(b, \lambda_t) &= (1 + \lambda_t)(\theta_{\star}^{\top} x_t) F_t^{\text{conv}}(b) - \lambda_t b F_t^{\text{conv}}(b), \\
\mathcal{L}_t^{\dagger}(b, \lambda_t) &= (1 + \lambda_t)(\theta_{\star}^{\top} x_t) F_t^{\dagger,\text{conv}}(b) - \lambda_t b F_t^{\dagger,\text{conv}}(b).
\end{aligned}
$$

Taking the absolute difference:

$$
\begin{aligned}
\left| \mathcal{L}_t(b) - \mathcal{L}_t^{\dagger}(b) \right| &= \left| [(1 + \lambda_t)(\theta_{\star}^{\top} x_t) - \lambda_t b] \cdot \left( F_t^{\text{conv}}(b) - F_t^{\dagger,\text{conv}}(b) \right) \right| \\
&\leq \left( (1 + \lambda_t)|\theta_{\star}^{\top} x_t| + \lambda_t |b| \right) \cdot \left| F_t^{\text{conv}}(b) - F_t^{\dagger,\text{conv}}(b) \right|.
\end{aligned}
$$

We bound the coefficients using $|\theta_{\star}^{\top} x_t| \leq 1$, $b \in [0,1]$, and $\lambda_t \leq \Lambda$:

$$
(1 + \lambda_t)|\theta_{\star}^{\top} x_t| + \lambda_t |b| \leq 1 + \Lambda + \Lambda = C_{\Lambda}.
$$

Substituting the convex envelope bound from Eq. (75):

$$
\left| \mathcal{L}_t(b) - \mathcal{L}_t^{\dagger}(b) \right| \leq C_{\Lambda} \cdot C\epsilon_t = O((1 + \Lambda)\epsilon_t).
\tag{76}
$$

**Step 3: Summation over Horizon.** Term (A1) evaluates this difference at $b_t^*$, and Term (A2) evaluates it at $\tilde{b}_t$. Since the bound derived in Step 2 is uniform over $b$, it applies to both terms equally.

$$(A1) + (A2) = \sum_{t=1}^{T} \left( \mathcal{L}_t(b_t^*) - \mathcal{L}_t^\dagger(b_t^*) \right) + \sum_{t=1}^{T} \left( \mathcal{L}_t^\dagger(\tilde{b}_t) - \mathcal{L}_t(\tilde{b}_t) \right)$$

$$\leq \sum_{t=1}^{T} 2 \cdot O((1 + \Lambda)\epsilon_t).$$

Since $\epsilon_t = \tilde{O}\left(\frac{1}{\sqrt{t}}\right)$, we have

$$(A1) + (A2) \leq O\left( (1 + \Lambda)\sqrt{T} \right) = \tilde{O}\left( (1 + \Lambda)\sqrt{T} \right).$$

$\square$

**Lemma C.3** (Expected RoS Violation Bound). *Assume the algorithm updates the dual variable $\lambda_{t+1} = \lambda_t \exp(-\eta \tilde{g}_t)$ with step size $\eta = 1/\sqrt{T}$ and $\lambda_1 = 1$. Assume the single-step estimation error satisfies $|g_t(b_t) - \tilde{g}_t(\tilde{b}_t)| \leq err_t$, and the normalized reward $\tilde{r}_t \in [0, 1]$. Then, the expected total RoS violation satisfies:*

$$\mathbb{E}\left[ \sum_{t=1}^{T} -g_t(b_t) \right] \leq \sqrt{T}\ln T + 2 + \mathbb{E}\left[ \sum_{t=1}^{T} err_t \right]. \tag{77}$$

*Proof.* The proof relies on the fact that the deterministic bound derived for the estimated violation holds for *any* realization of the algorithm's trajectory. We then apply the linearity of expectation.

Consider any specific realization of the process. $\tilde{b}_t$ maximizes the estimated Lagrangian $\mathcal{L}_t(b)$. Since $b = 0$ is feasible with zero value:

$$\mathcal{L}_t(\tilde{b}_t) = (1 + \lambda_t)\tilde{r}_t(\tilde{b}_t) - \lambda_t \tilde{c}_t(\tilde{b}_t) \geq 0. \tag{78}$$

Rearranging terms with $\tilde{g}_t = \tilde{r}_t - \tilde{c}_t$:

$$\lambda_t \tilde{g}_t \geq -\tilde{r}_t \implies -\tilde{g}_t \leq \frac{\tilde{r}_t}{\lambda_t}. \tag{79}$$

Since $\tilde{r}_t \leq 1$ and $\lambda_t \geq 1$, we have the deterministic single-step bound:

$$-\tilde{g}_t \leq \frac{1}{\lambda_t}. \tag{80}$$

This inequality holds with probability 1, contingent only on the optimality of $\tilde{b}_t$ and the definitions.

Let $V_{est} = \sum_{t=1}^{T} -\tilde{g}_t$ be the random variable representing the total estimated violation. We show that $V_{est}$ is bounded by a constant almost surely. Set a threshold $K = \frac{1}{\eta}\ln T = \sqrt{T}\ln T$. For any realization, we define a critical time step $T'$ (which is a random stopping time):

$$T' = \max \left\{ \tau \in [0, T] : \sum_{t=1}^{\tau} -\tilde{g}_t \leq K \right\}. \tag{81}$$

We split the sum into three parts based on this $T'$:

$$\sum_{t=1}^{T} -\tilde{g}_t = \sum_{t=1}^{T'} -\tilde{g}_t + (-\tilde{g}_{T'+1}) + \sum_{t=T'+2}^{T} -\tilde{g}_t. \tag{82}$$

- **Part 1:** By definition of $T'$, $\sum_{t=1}^{T'} -\tilde{g}_t \leq K$.

- **Part 2:** The single step violation is bounded: $-\tilde{g}_{T'+1} \leq 1$.

- **Part 3:** For $t \geq T' + 2$, the cumulative violation up to $t - 1$ strictly exceeds $K$. Thus, the dual variable satisfies:

$$\lambda_t = \exp\left(\eta \sum_{i=1}^{t-1} -\tilde{g}_i\right) > \exp(\eta K) = \exp(\ln T) = T.$$

Using Eq. (80), the violation is suppressed:

$$\sum_{t=T'+2}^{T} -\tilde{g}_t \leq \sum_{t=T'+2}^{T} \frac{1}{\lambda_t} < \sum_{t=T'+2}^{T} \frac{1}{T} \leq 1.$$

Combining these, for any realization, the total estimated violation is deterministically bounded:

$$\sum_{t=1}^{T} -\tilde{g}_t(\tilde{b}_t) \leq K + 1 + 1 = \sqrt{T} \ln T + 2. \tag{83}$$

Since this inequality holds almost surely, taking the expectation yields:

$$\mathbb{E}\left[\sum_{t=1}^{T} -\tilde{g}_t(\tilde{b}_t)\right] \leq \sqrt{T} \ln T + 2. \tag{84}$$

We relate the true violation to the estimated violation using the error term $\text{err}_t$:

$$-g_t(b_t) \leq -\tilde{g}_t(\tilde{b}_t) + |g_t(b_t) - \tilde{g}_t(\tilde{b}_t)| \leq -\tilde{g}_t(\tilde{b}_t) + \text{err}_t. \tag{85}$$

Summing over $t$ and taking the expectation on both sides:

$$\mathbb{E}\left[\sum_{t=1}^{T} -g_t(b_t)\right] \leq \mathbb{E}\left[\sum_{t=1}^{T} -\tilde{g}_t(\tilde{b}_t)\right] + \mathbb{E}\left[\sum_{t=1}^{T} \text{err}_t\right]$$

$$\leq \left(\sqrt{T} \ln T + 2\right) + \mathbb{E}\left[\sum_{t=1}^{T} \text{err}_t\right]. \tag{86}$$

$\square$

**Lemma C.4** (Expected Cumulative Estimation and Truncation Error). *Let $\text{err}_t = |g_t(b_t) - \tilde{g}_t(\tilde{b}_t)|$. Assuming the standard OFUL assumptions and the Oracle Assumption hold with a deterministic bound sequence $\epsilon_t$ (e.g., $\epsilon_t = O(t^{-1/2})$), the expected cumulative error is bounded by:*

$$\mathbb{E}\left[\sum_{t=1}^{T} err_t\right] \leq \tilde{O}\left(\sqrt{dT} + \sum_{t=1}^{T} \epsilon_t\right) = \tilde{O}(\sqrt{dT}). \tag{87}$$

*Proof.* By the triangle inequality and linearity of expectation:

$$\mathbb{E}\left[\sum_{t=1}^{T} \text{err}_t\right] \leq \underbrace{\mathbb{E}\left[\sum_{t=1}^{T} |g_t(b_t) - \tilde{g}_t(b_t)|\right]}_{(\text{I})} + \underbrace{\mathbb{E}\left[\sum_{t=1}^{T} |\tilde{g}_t(b_t) - \tilde{g}_t(\tilde{b}_t)|\right]}_{(\text{II})}.$$

**Part (I): Expected Estimation Error.** Let $\mathcal{E}$ be the high-probability "Good Event" (where parameters lie in ellipsoids and distribution error is bounded by $\epsilon_t$).

$$\mathbb{E}[\text{Error (I)}] = \mathbb{E}[\text{Error (I)} \mid \mathcal{E}]\mathbb{P}(\mathcal{E}) + \mathbb{E}[\text{Error (I)} \mid \mathcal{E}^c]\mathbb{P}(\mathcal{E}^c)$$

$$\leq \mathbb{E}\left[\sum_{t=1}^{T}(2w_t(b_t) + O(\epsilon_t)) \,\Big|\, \mathcal{E}\right] \cdot 1 + O(T) \cdot \frac{1}{T}$$

$$= 2\mathbb{E}\left[\sum_{t=1}^{T} w_t(b_t)\right] + O\left(\sum_{t=1}^{T} \epsilon_t\right) + O(1). \tag{88}$$

Note that $\sum \epsilon_t$ is now outside the expectation because $\epsilon_t$ is a deterministic bound sequence. Using the Weighted OFUL width bound $\mathbb{E}[\sum w_t] \leq \tilde{O}(\sqrt{dT})$ and $\sum_{t=1}^{T} t^{-1/2} \leq 2\sqrt{T}$:

$$\mathbb{E}[\text{Error (I)}] \leq \tilde{O}(\sqrt{dT}) + \tilde{O}(\sqrt{T}) = \tilde{O}(\sqrt{dT}).$$

**Part (II): Expected Truncation Error.** The truncation threshold $z_t \approx \beta_t \sqrt{d/T} + 4\epsilon_t$ is also a deterministic sequence (given fixed confidence schedules $\beta_t$).

$$
\begin{aligned}
\mathbb{E}\left[\sum_{t=1}^{T} \text{Error (II)}\right] &\leq \mathbb{E}\left[\sum_{t=1}^{T} L_g z_t\right] \\
&\leq \sum_{t=1}^{T} \tilde{O}\left(\sqrt{\frac{d}{T}} + \epsilon_t\right) \\
&= \tilde{O}(\sqrt{dT}) + \tilde{O}(\sqrt{T}).
\end{aligned}
\tag{89}
$$

*(Note: Since $z_t$ is deterministic in the algorithm design, the expectation only applies if there are random components in $\beta_t$, but usually $\beta_t$ is fixed effectively removing the randomness from this term as well).*

Combining both parts:

$$\mathbb{E}\left[\sum_{t=1}^{T} \text{err}_t\right] \leq \tilde{O}(\sqrt{dT}).$$

$\square$

## C.2. Proof of Lemma 5.2

We establish that the optimal dual variable $\lambda^\star$ is naturally bounded by the inverse of the Slater margin. This justifies the use of a projection step in the dual update.

*Proof.* Consider the hindsight convex optimization problem. Let $V^\star$ be the optimal cumulative reward of the primal problem subject to the cumulative RoS constraint. The Lagrangian function for a policy $\pi$ and a dual variable $\lambda \geq 0$ is:

$$
\begin{aligned}
\mathcal{L}(\pi, \lambda) &= \sum_{t=1}^{T} \mathbb{E}[r_t(\pi(x_t))] + \lambda \sum_{t=1}^{T} \mathbb{E}[g_t(\pi(x_t))] \\
&= \sum_{t=1}^{T} \mathbb{E}\left[(1+\lambda)r_t(\pi(x_t)) - \lambda c_t(\pi(x_t))\right].
\end{aligned}
\tag{90}
$$

(Note: This matches the Lagrangian form $\mathcal{L}_t$ used in our Regret Decomposition).

By Strong Duality (which holds as the problem is convex with Slater condition), the optimal value $V^\star$ satisfies:

$$V^\star = \max_\pi \mathcal{L}(\pi, \lambda^\star) \geq \mathcal{L}(\pi_{\text{slater}}, \lambda^\star), \tag{91}$$

where $\pi_{\text{slater}}$ is the policy satisfying the Slater condition margin. Expanding the Lagrangian for the Slater policy:

$$
\begin{aligned}
V^\star &\geq \sum_{t=1}^{T} \mathbb{E}[r_t(\pi_{\text{slater}}(x_t))] + \lambda^\star \sum_{t=1}^{T} \mathbb{E}[g_t(\pi_{\text{slater}}(x_t))] \\
&\geq \sum_{t=1}^{T} \mathbb{E}[r_t(\pi_{\text{slater}}(x_t))] + \lambda^\star \cdot (T\delta).
\end{aligned}
\tag{92}
$$

Rearranging to isolate $\lambda^\star$:

$$\lambda^\star(T\delta) \leq V^\star - \sum_{t=1}^{T} \mathbb{E}[r_t(\pi_{\text{slater}}(x_t))]. \tag{93}$$

We now bound the reward difference on the right-hand side. Since the uplift value is bounded by $|\theta_\star^\top x_t| \leq 1$ and $F_t(b) \in [0, 1]$, the reward $r_t(b) \in [0, 1]$. The cumulative difference is at most $T \cdot (1 - 0) = T$.

$$\lambda^\star(T\delta) \leq T. \tag{94}$$

Dividing by $T\delta$ yields the bound:

$$\lambda^\star \leq \frac{1}{\delta}. \tag{95}$$

$\square$

### C.3. Proof of Lemma 5.3 and Lemma 5.4

**Lemma C.5** (Concentration via Adaptive Martingales with Estimated Weights). *Let $T_0 = \sqrt{T}$. Under Assumption (Context Diversity & Overlap), even if $\{F_t\}$ is an adaptive adversarial sequence and weights are estimated using $\hat{F}_t$, with probability at least $1 - 4/T$:*

1. ***Exploration Guarantee:** The estimated weighted covariance matrix $\hat{A}_{T_0} = \sum_{t=1}^{T_0} \hat{F}_t(b_t)(1 - \hat{F}_t(b_t))x_t x_t^\top$ satisfies:*

$$\lambda_{\min}(\hat{A}_{T_0}) \geq \frac{\kappa T_0}{8}.$$

2. ***Estimation Error:** The parameter estimation error satisfies:*

$$\|\hat{\theta}_{T_0} - \theta_\star\|_2 \leq \tilde{O}(T^{-1/4}).$$

*Proof.* **Step 1: Lower Bound on Covariance (Perturbation Argument).** We first analyze the "ideal" covariance matrix $A_{T_0}$ constructed using true weights $w_t = F_t(b_t)(1 - F_t(b_t))$, and then bound the perturbation caused by using estimated weights $\hat{w}_t = \hat{F}_t(b_t)(1 - \hat{F}_t(b_t))$.

**(a) Ideal Matrix Lower Bound:** Define $Z_t = w_t x_t x_t^\top$. As established in previous analyses for the ideal case, the random bidding policy $\pi_{\text{uni}}$ ensures $\mathbb{E}[Z_t \mid \mathcal{F}_{t-1}] \succeq c_{\text{var}}\kappa_x I_d$. Applying the Matrix Chernoff bound for adaptive sequences, with high probability:

$$\lambda_{\min}(A_{T_0}) \geq \frac{\kappa T_0}{4}. \tag{96}$$

**(b) Bounding the Perturbation:** The algorithm computes $\hat{A}_{T_0}$ using $\hat{w}_t$. Define the scalar function $f(u) = u(1 - u)$. Since $f'(u) = 1 - 2u$, we have $|f'(u)| \leq 1$ for $u \in [0, 1]$. By the Lipschitz property and Lemma 3.1:

$$|\hat{w}_t - w_t| = |f(\hat{F}_t(b_t)) - f(F_t(b_t))| \leq |\hat{F}_t(b_t) - F_t(b_t)| \leq \epsilon_t. \tag{97}$$

The spectral norm difference between the estimated and true matrices is:

$$\|\hat{A}_{T_0} - A_{T_0}\|_{\text{op}} = \left\|\sum_{t=1}^{T_0}(\hat{w}_t - w_t)x_t x_t^\top\right\|_{\text{op}} \leq \sum_{t=1}^{T_0}|\hat{w}_t - w_t| \cdot \|x_t\|_2^2$$

$$\leq \sum_{t=1}^{T_0}\epsilon_t = \sum_{t=1}^{T_0}\tilde{O}\left(\sqrt{\frac{d}{t}} + \|x_t\|_{\Sigma_t^{-1}}\right). \tag{98}$$

In the burn-in phase, since we bid uniformly, the unweighted covariance $\Sigma_t$ grows linearly: $\lambda_{\min}(\Sigma_t) \gtrsim t$. Thus, $\|x_t\|_{\Sigma_t^{-1}} \leq 1/\sqrt{\lambda_{\min}(\Sigma_t)} = O(1/\sqrt{t})$. Summing the error terms:

$$\|\hat{A}_{T_0} - A_{T_0}\|_{\text{op}} \leq \tilde{O}\left(\sum_{t=1}^{T_0}\frac{\sqrt{d}}{\sqrt{t}}\right) = \tilde{O}(\sqrt{dT_0}). \tag{99}$$

**(c) Conclusion for Covariance:** Using Weyl's inequality $\lambda_{\min}(\hat{A}) \geq \lambda_{\min}(A) - \|\hat{A} - A\|_{\text{op}}$:

$$\lambda_{\min}(\hat{A}_{T_0}) \geq \frac{\kappa T_0}{4} - \tilde{O}(\sqrt{dT_0}). \tag{100}$$

For sufficiently large $T$ (and thus $T_0$), the linear term dominates the square root term. Specifically, for $T_0$ large enough, $\tilde{O}(\sqrt{dT_0}) \leq \frac{\kappa T_0}{8}$. Thus:

$$\lambda_{\min}(\hat{A}_{T_0}) \geq \frac{\kappa T_0}{8}.$$

**Step 2: Upper Bound on Estimation Error.** The estimator $\hat{\theta}_{T_0}$ is the Weighted Least Squares solution using estimated weights. Applying the standard Self-Normalized Martingale Concentration inequality (Abbasi-Yadkori et al., 2011), we have with high probability:

$$\|\hat{\theta}_{T_0} - \theta_\star\|_{\hat{A}_{T_0}} \leq \beta_{T_0} = \tilde{O}(\sqrt{d}).$$

Converting to Euclidean norm using the eigenvalue bound from Step 1:

$$\|\hat{\theta}_{T_0} - \theta_\star\|_2 \leq \frac{\beta_{T_0}}{\sqrt{\lambda_{\min}(\hat{A}_{T_0})}} \leq \frac{\tilde{O}(\sqrt{d})}{\sqrt{\kappa T_0/8}} = \tilde{O}(T_0^{-1/2}).$$

Substituting $T_0 = \sqrt{T}$, we get the final rate:

$$\|\hat{\theta}_{T_0} - \theta_\star\|_2 \leq \tilde{O}(T^{-1/4}).$$

$\square$

**Lemma C.6** (Validity of Projection Interval). *Suppose the high-probability event in Lemma 5.3 holds (i.e., $\|\hat{\theta}_{T_0} - \theta_\star\|_2 \leq \tilde{O}(T^{-1/4})$). For a sufficiently large horizon $T$ (specifically $T \geq poly(1/\delta_{true})$), the projection bound $\Lambda = 2/\hat{\delta}$ constructed from the estimated parameter $\hat{\theta}_{T_0}$ satisfies:*

$$\lambda^\star \in [0, \Lambda], \tag{101}$$

*where $\lambda^\star$ is the optimal dual multiplier for the hindsight problem.*

*Proof.* Let $\delta(\theta)$ denote the Slater margin associated with parameter $\theta$. From Lemma 5.3, we have the estimation error bound $\|\hat{\theta}_{T_0} - \theta_\star\|_2 \leq \epsilon_{est}$ with $\epsilon_{est} = \tilde{O}(T^{-1/4})$. By the Lipschitz continuity of the Slater margin function (as established in Assumption 2.6 analysis), we have:

$$|\delta(\hat{\theta}_{T_0}) - \delta(\theta_\star)| \leq \|\hat{\theta}_{T_0} - \theta_\star\|_2 \leq \epsilon_{est}.$$

This implies the estimated margin is bounded by $\hat{\delta} \leq \delta_{true} + \epsilon_{est}$. The condition $T$ is sufficiently large ensures that the estimation error is small relative to the true margin, specifically $\epsilon_{est} \leq \delta_{true}$. Thus, $\hat{\delta} \leq 2\delta_{true}$. Substituting this into the definition of $\Lambda$:

$$\Lambda = \frac{2}{\hat{\delta}} \geq \frac{2}{2\delta_{true}} = \frac{1}{\delta_{true}}. \tag{102}$$

Since standard duality theory bounds the optimal dual variable by the inverse of the Slater margin ($\lambda^\star \leq 1/\delta_{true}$), we conclude $\lambda^\star \leq \Lambda$. $\square$

## Auxiliary Lemmas

**Lemma C.7** (Lagrangian Discretization Error). *Suppose Assumptions 2.1–2.3 hold. Let $b^* = \arg\max_{b \in [0,1]} J_t(b)$ be the optimal continuous bid maximizing the true expected Lagrangian $J_t(b) = \mathbb{E}[r_t(b)] - Z\mu_t \mathbb{E}[c_t(b)]$. Let $b_{grid} \in \mathcal{B}_K$ be the closest discretized bid in the grid such that $|b_{grid} - b^*| \leq \frac{1}{2K}$.*

*The discretization gap for the true expected Lagrangian is bounded by:*

$$|J_t(b_{grid}) - J_t(b^*)| \leq \frac{L+1}{2K}(1 + Z\mu_t) \tag{103}$$

*Proof.* The true expected Lagrangian is defined as:

$$J_t(b) = G_t(b)(\theta_\star^\top x_t - b) + (1 - G_t(b))v_{t,0} - Z\mu_t \cdot bG_t(b)$$

We analyze the Lipschitz continuity of $J_t(b)$ with respect to $b$. Recall that by Assumption 2.3, $G_t(b)$ is $L$-Lipschitz (density bounded by $L$). The terms involving $b$ are products of Lipschitz functions and bounded terms.

Specifically, the derivative (or subgradient) magnitude can be bounded as:

$$\left|\frac{d}{db}\mathbb{E}[r_t(b)]\right| = \left|g_t(b)(\theta_\star^\top x_t - b) - G_t(b) - g_t(b)v_{t,0}\right|$$

$$\leq L(1-0) + 1 + L(1) = 2L + 1 \quad \text{(Loose bound)}$$

However, a tighter bound used in the lemma statement can be derived by grouping:

$$\mathbb{E}[r_t(b)] = G_t(b)(\text{Uplift} - b) + v_{t,0}$$

The derivative is $g_t(b)(\text{Uplift} - b) - G_t(b)$. Since $\text{Uplift} \in [-1, 1]$ and $b \in [0, 1]$, the Lipschitz constant is bounded by $L + 1$.

For the cost term $\mathbb{E}[c_t(b)] = bG_t(b)$:

$$\left|\frac{d}{db}(bG_t(b))\right| = |G_t(b) + bg_t(b)| \leq 1 + L$$

Combining these, the total Lipschitz constant of $J_t(b)$ is bounded by:

$$L_{total} \leq (L+1) + Z\mu_t(L+1) = (L+1)(1 + Z\mu_t)$$

Since $|b_{grid} - b^*| \leq \frac{1}{2K}$, the value difference is:

$$|J_t(b_{grid}) - J_t(b^*)| \leq L_{total}|b_{grid} - b^*| \leq \frac{L+1}{2K}(1 + Z\mu_t)$$

$\square$

# D. Technical Lemmas

**Lemma D.1** (Wen et al. (2025)). *With probability at least $1 - T^{-2}$, for all $t \in [T]$:*

$$|\hat{\theta}_t^\top x_t - \theta_\star^\top x_t| \leq \beta_t \|x_t\|_{A_t^{-1}} . \tag{104}$$

**Lemma D.2** (Dual Regret Bound). *Assume the dual variable $\mu_t$ is updated via Online Mirror Descent with a step size $\eta \propto 1/\sqrt{T}$. Regardless of the specific linear structure, as long as the consumption $c_t(b_t) \in [0, 1]$, we have the following lower bound on the dual gain:*

$$\sum_{t=1}^{\tau-1} \mu_t\left(c_t(b_t) - \frac{B}{T}\right) \geq -O(\sqrt{T}) + B\left(1 - \frac{\tau-1}{T}\right) - 1. \tag{105}$$

*Proof.* We assume that $\mu_t$ is updated via Online Mirror Descent (OMD) with a projection step $\mu_{t+1} = \Pi_{[0,1]}(\mu_{t+1}^{\text{raw}})$.

- Define the "gain" of the dual loss function as $f_t(\mu) = \mu \cdot \left(c_t(b_t) - \frac{B}{T}\right)$.

- The algorithm aims to maximize the cumulative dual gain.

According to the standard regret bound of OMD, the regret is bounded by the sum of Bregman divergences. Crucially, since we assume the optimal dual variable satisfies $\mu^* \in [0, 1]$ (Assumption 2.5), the projection of $\mu_{t+1}$ onto $[0, 1]$ ensures that the distance to the optimum does not increase: $D(\mu^*||\mu_{t+1}) \leq D(\mu^*||\mu_{t+1}^{\text{raw}})$. Thus, the standard regret bound holds for any fixed $\mu \in [0, 1]$:

$$\sum_{t=1}^{\tau-1} f_t(\mu_t) \geq \sum_{t=1}^{\tau-1} f_t(\mu) - \mathcal{R}_{\text{dual}}(\tau - 1)$$

where $\mathcal{R}_{\text{dual}}(\tau)$ is typically of order $O(\sqrt{\tau})$. Substituting our specific form:

$$\sum_{t=1}^{\tau-1} \mu_t \left( c_t(b_t) - \frac{B}{T} \right) \geq \max_{\mu \in [0,1]} \left[ \mu \cdot \left( \sum_{t=1}^{\tau-1} c_t(b_t) - \frac{\tau-1}{T} B \right) \right] - O(\sqrt{T}) \tag{106}$$

We need to analyze the maximization term on the right-hand side. This depends on whether the auction process ends early due to "budget exhaustion" or completes the full horizon.

**Case 1: Early Stopping ($\tau \leq T$).** $\tau$ is the first time index where the budget is insufficient for the payment $b_\tau$. This implies that the consumption in the first $\tau - 1$ rounds satisfies the budget constraint, but attempting the $\tau$-th round would exceed the budget. Since the normalized bid $b_t \leq 1$, the remaining budget at $\tau$ must be less than 1. Mathematically:

$$B - \sum_{t=1}^{\tau-1} c_t(b_t) < b_\tau \leq 1 \implies \sum_{t=1}^{\tau-1} c_t(b_t) > B - 1$$

At this point, we examine the cumulative term of Term B over the first $\tau - 1$ rounds. Let $\Delta_{\text{budget}}$ be the total deviation from the target budget consumption:

$$\Delta_{\text{budget}} = \sum_{t=1}^{\tau-1} c_t(b_t) - \frac{\tau-1}{T} B$$

Substituting the lower bound of total consumption:

$$\Delta_{\text{budget}} > (B - 1) - \frac{\tau-1}{T} B = B \left( 1 - \frac{\tau-1}{T} \right) - 1$$

Since $\tau \leq T$, the term $1 - \frac{\tau-1}{T}$ is positive. To maximize the expression $\mu \cdot \Delta_{\text{budget}}$ in Equation (106), the optimal fixed dual variable is $\mu^* = 1$. Thus:

$$\sum_{t=1}^{\tau-1} \mu_t \left( c_t(b_t) - \frac{B}{T} \right) \geq 1 \cdot \left[ B \left( 1 - \frac{\tau-1}{T} \right) - 1 \right] - O(\sqrt{T})$$

$$= -1 + B \left( 1 - \frac{\tau-1}{T} \right) - O(\sqrt{T}).$$

This establishes the claim for the early stopping case.

**Case 2: Full Horizon ($\tau = T + 1$).** In this case, the algorithm completes $T$ rounds of bidding (i.e., $\tau - 1 = T$), and the total consumption satisfies the constraint: $\sum_{t=1}^{T} c_t(b_t) \leq B$. The term inside the maximization becomes:

$$\sum_{t=1}^{T} c_t(b_t) - B \leq 0$$

To maximize $\mu \cdot (\text{non-positive value})$ where $\mu \in [0, 1]$, the optimal dual variable is $\mu^* = 0$. The maximum value is then 0. Therefore:

$$\sum_{t=1}^{T} \mu_t \left( c_t(b_t) - \frac{B}{T} \right) \geq 0 - O(\sqrt{T})$$

Note that in this case, $1 - \frac{\tau-1}{T} = 0$, so the term $B(1 - \frac{\tau-1}{T})$ vanishes. The inequality holds trivially as $-O(\sqrt{T}) \geq -1 - O(\sqrt{T})$. $\qquad \square$

**Lemma D.3** (Wen et al. (2025)). *Let $G$ be a continuous CDF on $[0, 1]$, $v \in [0, 1]$, and $z \in (0, \frac{1}{2}]$. Let $\hat{G}$ be another CDF satisfying $\|G - \hat{G}\|_\infty \leq \epsilon$. For any bid $b \in [0, 1]$, define the truncated bid $b'$ as:*

$$b' = \min \left\{ \max\{b, \hat{G}^{-1}(z)\}, \hat{G}^{-1}(1 - z) \right\}.$$

*Then, the estimated probability satisfies $z \leq \hat{G}(b') \leq 1 - z + 2\epsilon$, and the potential reward loss is bounded by:*

$$\hat{G}(b)(v - b) - \hat{G}(b')(v - b') \leq z + 2\epsilon.$$

**Lemma D.4** (Elliptical Potential Lemma). *For any given sequence of vectors $\{z_\tau\}_{\tau=1}^{t-1} \in \mathbb{R}^d$ satisfying $\|z_\tau\|_2 \leq 1$, let the Gram matrix be $A_s = \lambda I + \sum_{\tau < s} z_\tau z_\tau^\top$ with $\lambda > 0$, for each $1 \leq s \leq t$. Then the following holds:*

$$\sum_{\tau < t} \|z_\tau\|_{A_\tau^{-1}}^2 \leq 2d \log\left(\lambda + \frac{t-1}{d}\right) + 2\log(\lambda).$$

*In particular, by the Cauchy-Schwarz inequality:*

$$\sum_{\tau < t} \|z_\tau\|_{A_\tau^{-1}} \leq \sqrt{2d(t-1)\log\left(1 + \frac{t-1}{d\lambda}\right)} + \sqrt{2(t-1)\log(\lambda)}.$$

**Lemma D.5** (Variance Proxy Bound via Truncation). *In the OFUL-FPA algorithm, with the truncation threshold $z = \min\left\{\beta_t\sqrt{\frac{d}{T}} + 4\epsilon_t, \frac{1}{2}\right\}$ and assuming $\epsilon_t \leq 0.1$, the variance proxy satisfies:*

$$\sigma_t(b_t)^2 \leq \frac{2}{\beta_t}\sqrt{\frac{T}{d}} + 5. \tag{107}$$

*Proof.* Let $z_{\text{base}} = \beta_t\sqrt{\frac{d}{T}}$. The threshold is $z = \min\{z_{\text{base}} + 4\epsilon_t, \frac{1}{2}\}$. As established in Lemma D.3, the estimated probability lies in the range $[z, 1 - z + 2\epsilon_t]$. The variance proxy is bounded by the inverse of the minimum variance in this interval.

**Case 1: Active Truncation ($z_{\text{base}} + 4\epsilon_t \leq \frac{1}{2}$)**   In this case, $z = z_{\text{base}} + 4\epsilon_t$. The distance from the boundary $\{0, 1\}$ is at least $d_R = z - 2\epsilon_t = z_{\text{base}} + 2\epsilon_t$. The variance proxy is bounded by:

$$\sigma_t^2 \leq \frac{1}{(z_{\text{base}} + 2\epsilon_t)(1 - (z_{\text{base}} + 2\epsilon_t))}.$$

Since $z_{\text{base}} + 4\epsilon_t \leq \frac{1}{2}$, we know $z_{\text{base}} + 2\epsilon_t < \frac{1}{2}$. This implies the term $(1 - (z_{\text{base}} + 2\epsilon_t)) > \frac{1}{2}$. Thus:

$$\sigma_t^2 \leq \frac{1}{(z_{\text{base}} + 2\epsilon_t) \cdot \frac{1}{2}} = \frac{2}{z_{\text{base}} + 2\epsilon_t} \leq \frac{2}{z_{\text{base}}} = \frac{2}{\beta_t}\sqrt{\frac{T}{d}}.$$

**Case 2: Saturated Truncation ($z_{\text{base}} + 4\epsilon_t > \frac{1}{2}$)**   In this case, the truncation hits the cap, so $z = \frac{1}{2}$. The probability range becomes $[\frac{1}{2}, \frac{1}{2} + 2\epsilon_t]$. The worst-case variance proxy occurs at the endpoint furthest from $\frac{1}{2}$, which is $p = \frac{1}{2} + 2\epsilon_t$.

$$\sigma_t^2 \leq \frac{1}{(\frac{1}{2} + 2\epsilon_t)(1 - (\frac{1}{2} + 2\epsilon_t))} = \frac{1}{(\frac{1}{2} + 2\epsilon_t)(\frac{1}{2} - 2\epsilon_t)} = \frac{1}{0.25 - 4\epsilon_t^2}.$$

Now we utilize the assumption $\epsilon_t \leq 0.1$. This implies $\epsilon_t^2 \leq 0.01$ and $4\epsilon_t^2 \leq 0.04$. The denominator is bounded from below:

$$0.25 - 4\epsilon_t^2 \geq 0.25 - 0.04 = 0.21.$$

Thus, the variance proxy is bounded by a constant:

$$\sigma_t^2 \leq \frac{1}{0.21} \approx 4.76 < 5.$$

**Conclusion**   Combining both cases, the variance proxy is always bounded by the sum of the two bounds (since one term is dominant in one regime and the other is effectively zero or subsumed):

$$\sigma_t(b_t)^2 \leq \max\left(5, \frac{2}{\beta_t}\sqrt{\frac{T}{d}}\right) \leq \frac{2}{\beta_t}\sqrt{\frac{T}{d}} + 5.$$

$\square$

**Lemma D.6** (Unified Lagrangian Discretization Error). *Suppose Assumptions 2.1–2.3 hold, where the winning probability $F_t(b)$ admits a density $f_t(b)$ bounded by L. Consider a generalized expected Lagrangian of the form:*

$$J_t(b) = \alpha_t \mathbb{E}[r_t(b)] - \beta_t \mathbb{E}[c_t(b)]$$

*where $\alpha_t, \beta_t \geq 0$ are dual-related weights. Let $b^* = \arg\max_{b \in [0,1]} J_t(b)$ be the optimal continuous bid, and let $b_{grid} \in \mathcal{B}_K$ be the closest discretized bid such that $|b_{grid} - b^*| \leq \frac{1}{2K}$.*

*The discretization gap is bounded by:*

$$|J_t(b_{grid}) - J_t(b^*)| \leq \frac{L+1}{2K}(\alpha_t + \beta_t) \tag{108}$$

*Proof.* The proof proceeds by establishing the Lipschitz continuity of the expected reward and cost functions with respect to the bid $b$, and then combining them for the Lagrangian.

**Step 1: Lipschitz Constants of Component Functions.** We analyze the sensitivity of the expected cost and reward to changes in the bid $b$. By Assumption 2.3, the PDF of the winning probability is bounded, i.e., $f_t(b) \in [0, L]$ for all $b \in [0, 1]$.

*1. Expected Cost $\mathbb{E}[c_t(b)]$.* In a First-Price Auction, the cost is the bid amount if the bidder wins, and 0 otherwise:

$$\mathbb{E}[c_t(b)] = b \cdot F_t(b).$$

Differentiating with respect to $b$:

$$\frac{d}{db}\mathbb{E}[c_t(b)] = 1 \cdot F_t(b) + b \cdot f_t(b).$$

Using the bounds $F_t(b) \in [0, 1]$, $b \in [0, 1]$, and $|f_t(b)| \leq L$, we bound the magnitude of the derivative:

$$\left|\frac{d}{db}\mathbb{E}[c_t(b)]\right| \leq |F_t(b)| + |b| \cdot |f_t(b)| \leq 1 + 1 \cdot L = L + 1.$$

*2. Expected Primal Reward (Surplus) $\mathbb{E}[r_t(b)]$.* In the Linear Treatment Effect (LTE) setting, the agent receives potential outcome $Y_t(1)$ if they win (treated) and $Y_t(0)$ if they lose (control). The expected surplus is:

$$\mathbb{E}[r_t(b)] = \mathbb{E}[Y_t(1) - b]F_t(b) + \mathbb{E}[Y_t(0)](1 - F_t(b))$$
$$= (\mathbb{E}[Y_t(1)] - b)F_t(b) + \mathbb{E}[Y_t(0)] - \mathbb{E}[Y_t(0)]F_t(b)$$
$$= \underbrace{(\mathbb{E}[Y_t(1)] - \mathbb{E}[Y_t(0)])}_{\text{Uplift } \tau_t} F_t(b) - bF_t(b) + \mathbb{E}[Y_t(0)].$$

Simplifying, we have $\mathbb{E}[r_t(b)] = (\tau_t - b)F_t(b) + \text{const}$, where $\tau_t = \theta_\star^\top x_t$ is the Conditional Average Treatment Effect (CATE). Differentiating with respect to $b$:

$$\frac{d}{db}\mathbb{E}[r_t(b)] = (-1) \cdot F_t(b) + (\tau_t - b) \cdot f_t(b).$$

Assuming normalized rewards such that the CATE and bid satisfy $|\tau_t - b| \leq 1$ (conservatively bounding the interaction range), we have:

$$\left|\frac{d}{db}\mathbb{E}[r_t(b)]\right| \leq |-F_t(b)| + |\tau_t - b| \cdot |f_t(b)| \leq 1 + 1 \cdot L = L + 1.$$

**Step 2: Lagrangian Discretization Error.** The generalized Lagrangian is defined as $J_t(b) = \alpha_t \mathbb{E}[r_t(b)] - \beta_t \mathbb{E}[c_t(b)]$. Using the linearity of the derivative and the triangle inequality:

$$\left|\frac{d}{db}J_t(b)\right| = \left|\alpha_t\frac{d}{db}\mathbb{E}[r_t(b)] - \beta_t\frac{d}{db}\mathbb{E}[c_t(b)]\right|$$
$$\leq \alpha_t\left|\frac{d}{db}\mathbb{E}[r_t(b)]\right| + \beta_t\left|\frac{d}{db}\mathbb{E}[c_t(b)]\right|$$
$$\leq \alpha_t(L + 1) + \beta_t(L + 1)$$
$$= (L + 1)(\alpha_t + \beta_t).$$

Let $L_{\text{total}} = (L+1)(\alpha_t + \beta_t)$ be the Lipschitz constant of $J_t(b)$. Since $b_{grid}$ is the closest point in the grid to the optimal continuous bid $b^*$, we have $|b_{grid} - b^*| \leq \frac{1}{2K}$. The discretization error is bounded by:

$$|J_t(b_{grid}) - J_t(b^*)| \leq L_{\text{total}}|b_{grid} - b^*| \leq \frac{L+1}{2K}(\alpha_t + \beta_t).$$

$\square$

**Lemma D.7** (Cumulative Width Bound for Weighted OFUL). *Consider a sequence of context vectors $\{x_t\}_{t=1}^T \subset \mathbb{R}^d$ with $\|x_t\|_2 \leq 1$ and a sequence of variance proxies $\{\hat{V}_t\}_{t=1}^T$ (where $\hat{V}_t \in [0, 1/4]$). Let the weighted covariance matrix be $A_t = \lambda I + \sum_{\tau=1}^t \hat{V}_\tau x_\tau x_\tau^\top$. Define the confidence width at round $t$ as:*

$$w_t = \beta_t \left( \hat{V}_t \|x_t\|_{A_{t-1}^{-1}} + C\|x_t\|_{A_{t-1}^{-1}}^2 \right), \tag{109}$$

*where $\beta_t$ is a non-decreasing sequence and $C > 0$ is a constant. Suppose the variance proxy is lower bounded via truncation such that its inverse $\sigma_t^2 = 1/\hat{V}_t$ satisfies:*

$$\sigma_t^2 \leq \frac{K}{\beta_t}\sqrt{T} + \sigma_{\max}^2, \tag{110}$$

*for some constants $K, \sigma_{\max} > 0$. Then, the cumulative sum of widths is bounded by:*

$$\sum_{t=1}^T w_t \leq \tilde{O}\left( \beta_T \sqrt{dT} \right). \tag{111}$$

**Lemma D.8** (Wen et al. (2025)). *Let $x_1, \ldots, x_n \in \mathbb{R}^d$ be vectors satisfying $\|x_i\|_2 \leq 1$ for all $i \in [n]$, and $\Sigma := 18I + \sum_{i=1}^n x_i x_i^\top$. Then there exists a subset $S \subseteq [n]$ such that $|S| \leq \frac{n}{2}$ and*

$$18I + \sum_{i \in S} x_i x_i^\top \geq \frac{\Sigma}{9}.$$

*Proof.* The proof uses a profound result from the Kadison-Singer problem. Let $y_i := \Sigma^{-1/2} x_i$ for each $i \in [n]$, so $\sum_{i=1}^n y_i y_i^\top \leq I$. Furthermore,

$$\|y_i\|_2^2 \leq \frac{1}{18}\|x_i\|_2^2 \leq \frac{1}{18}.$$

By Marcus et al. (2015, Corollary 1.5) and taking $r = 2, \delta = \frac{1}{18}$), there exists a partition $\{S_1, S_2\}$ of $[n]$ such that for $j \in \{1, 2\}$,

$$\sum_{i \in S_j} y_i y_i^\top \leq \left( \frac{1}{\sqrt{r}} + \sqrt{\delta} \right)^2 I = \frac{8}{9}I \implies \sum_{i \in S_j} x_i x_i^\top \leq \frac{8}{9}\Sigma.$$

Without loss of generality, assume $|S_1| \geq \frac{n}{2}$, then choosing $S = S_2$ satisfies $|S| \leq \frac{n}{2}$ and

$$18I + \sum_{i \in S} x_i x_i^\top = \Sigma - \sum_{i \in S_1} x_i x_i^\top \geq \frac{\Sigma}{9}.$$

This proves the lemma. $\square$

