# OpenReview forum: "Learning to Bid with Unknown Private Values in Budget-Constrained First-Price Auctions"
_ICML.cc/2026/Conference — Submitted to ICML 2026_

### Official Review · Reviewer_bwvR · 2026-03-03

**Soundness:** 3
**Presentation:** 3
**Significance:** 3
**Originality:** 2
**Overall Recommendation:** 4
**Confidence:** 3

**Summary:**

This paper studies first-price auctions with a linear treatment effect model with budget or return on spend constraints. Especially, this paper proposes a unified primal-dual framework for constrained FPAs that jointly learns the latent LTE valuation parameters and competitiors bid distribution. To solve this problem, they leverage a strong Slater condition and a novel adaptive burn-in procedure to stabilize the dual variables. Their methods achieves near-optmal regret guarantees.

**Compliance With Llm Reviewing Policy:**

Affirmed.

**Final Justification:**

I will maintain my score, but I still have a concern that the algorithm's performance is highly dependent on the assumption of the strong Slater condition.

**Key Questions For Authors:**

Please see the weaknesses.

**Limitations:**

No.
Could you provide details about the assumption for the strong Slater condition? Could you provide a justification for this condition?

**Strengths And Weaknesses:**

Strengths
1. This paper considers a practical scenario where LTE valuation parameters and competitors' bid distribution are unkown and they are linearly dependent on the context.
2. To address the challenges, they utilize an adaptive burn-in phase to esitmate slater constant and split sample CDF estimation.


Weaknesses
1. The main concern is that the algorithmic and theoretical contribution is somewhat reconstructin gthe previous result rather than a novel.
2. Another concern is that the algorithm's performance is highly dependent on the assumption of the strong Slater condition.
3. The last one is that it assumes full information feedback for $m_t$

---

> ### Author Rebuttal · Authors · 2026-03-31
>
> We deeply appreciate the reviewer's positive evaluation and insightful comments. We address questions from the reviewer as follows.
>
> **The main concern is that the algorithmic and theoretical contribution is somewhat reconstructing the previous result rather than a novel.**
>
> We sincerely thank the reviewer for their feedback. We respectfully clarify that our work is not a trivial combination of existing methods; a naive synthesis of these domains mathematically fails and is computationally intractable. We overcome these barriers through three core innovations:
>
> 1. Realistic "Doubly Linear" Formulation: Prior works isolate Linear Treatment Effects (LTE) for uplift or linear models for bids. However, in Real-Time Bidding, observable contexts (e.g., user demographics) signal value to both us *and* our competitors. We provide the first optimal $\tilde{O}(d\sqrt{T})$ regret analysis where both value uplift and market price jointly depend on the same context.
>
> 2. Bypassing FNP-Hardness via Sample Splitting: Prior LTE auction frameworks (e.g., Wen et al.) rely on the Marcus-Spielman-Srivastava (MSS) theorem for joint CDF estimation. Because finding the exact MSS partition is FNP-hard, this only provides a non-constructive existence proof. We bridge this gap with a novel *split-sample estimation procedure* (Algorithm 1) that uses strategic regularization to absorb the errors of random sampling, delivering a computationally efficient online algorithm.
>
> 3. Taming the Explosive RoS Cross-Term: Applying standard Primal-Dual frameworks to the "doubly linear" RoS setting introduces a fatal mathematical flaw: the estimation error ($w_t$) infects the constraint evaluation, creating an explosive cross-term $\sum \lambda_t w_t$. If the dual multiplier $\lambda_t$ grows arbitrarily, regret becomes unbounded. We resolve this by introducing the Strong Slater Condition and an adaptive burn-in phase, which guarantees a bounded optimal dual variable ($\lambda^\star \le 1/\delta$) and stabilizes the regret.
>
> We have updated the manuscript (specifically after Lemma 3.2 and near Assumption 6) to make these technical departures and justifications explicitly clear to the reader.
>
> **Could you provide details about the assumption for the strong Slater condition? Could you provide a justification for this condition?**
>
> We thank the reviewer for highlighting this. The Strong Slater Condition (which requires that some baseline strategy yields a net profit margin of at least $\delta$ per impression on average, i.e., $\frac{1}{T} \sum_{t=1}^T \mathbb{E}[g_t(\pi_{\text{slater}}(x_t))] \ge \delta$) is a critical pillar of our RoS-constrained analysis. We will add a dedicated paragraph in the revision summarizing its necessity:
>
> 1. Technical Necessity (Bounding the Novel Cross-Term): Unlike standard settings with fully observable values, our "doubly linear" RoS framework jointly estimates hidden value uplifts and opponent bids. Because the dual multiplier $\lambda_t$ scales the estimated constraint, the estimation error (confidence radius $w_t$) infects both the reward and constraint evaluations. This creates an explosive cross-term in the regret
> $$\sum_{t=1}^T \lambda_t w_t(\tilde{b}_t).$$
> If $\lambda_t$ grows arbitrarily, this term destroys the sublinear $\tilde{O}(\sqrt{T})$ regret bound. The Slater Condition guarantees a strictly positive margin $\delta$, which bounds the optimal dual variable via standard duality theory ($\lambda^\star \le 1/\delta$). This allows us to safely cap the dual space and tame the error to $(1+\Lambda) \sum w_t$.
>
> 2. Practical Justification (Feasibility of Target RoS): In real-world digital advertising, this condition simply means the advertiser's Target RoS is genuinely achievable. It assumes at least one conservative baseline strategy can strictly satisfy the RoS goal. If no such policy existed, the target would be too aggressive for the market, making the problem mathematically ill-posed. Because advertisers set RoS targets based on historical baseline campaigns, the Slater condition merely formalizes that the algorithm has been given a feasible financial guardrail.
>
> **The last one is that it assumes full information feedback for $m_t$?**
>
> Thanks for pointing this out. In the initial submission, we have only included the full information feedback setting. But during the rebuttal phase, we found that we can generalize our proof technology to the case of winning-bid feedback, due to Assumption 2.4 (Gaussian noise). Please refer to the response to reviewer 9i55 on "The model assumes that the highest competing bid is observed even when losing..." for details.

---

> > ### Author Rebuttal · Reviewer_bwvR · 2026-04-03
> >
> > Thank you for your detailed response. I will keep my score.

---

### Official Review · Reviewer_9i55 · 2026-03-03

**Soundness:** 3
**Presentation:** 3
**Significance:** 3
**Originality:** 3
**Overall Recommendation:** 3
**Confidence:** 4

**Summary:**

This paper studies repeated bidding in first-price auctions when the bidder does not know the true private value of each impression. Instead of observing value directly, the bidder models value as uplift, meaning the difference between two potential outcomes. So value has to be learned over time from partial feedback.

Each round, the bidder observes a context vector. Both the uplift value and the highest competing bid are assumed to follow linear models in that context (plus noise). The bidder faces three settings: no constraint, a hard budget constraint, and a return-on-spend constraint. The goal is to learn to bid well while respecting these constraints and keeping regret low.

A key modeling assumption is that the highest competing bid is observed every round, even if the bidder loses. This is important because it lets the learner estimate the market price distribution directly.

The proposed algorithm combines linear confidence bounds for learning the uplift parameter with a primal-dual update for handling constraints. For return-on-spend, the paper adds a burn-in exploration phase to estimate a feasibility slack, which is then used to keep the dual variable under control. The main theoretical claim is that regret scales on the order of square root of time (up to logs) in all three settings, under the stated assumptions.

**Compliance With Llm Reviewing Policy:**

Affirmed.

**Key Questions For Authors:**

The model assumes that the highest competing bid is observed even when losing. Honestly, how essential is this for the regret proof? If we only observe the price upon winning, does the analysis fundamentally break?

In the return-on-spend case, how does the regret bound behave as the feasibility slack gets small? Is the dependence explicit anywhere, or does it effectively hide inside constants?

The overlap-type condition ensures enough variation for stable estimation. In practice, do you expect this to hold automatically under your bidding policy, or does it require careful tuning?

**Limitations:**

The guarantees rely on several strong structural assumptions: full observation of market prices, linear models for value and competition, an overlap condition, and a nontrivial feasibility slack for return-on-spend. These are reasonable in a stylized model, but they narrow the scope of the results. If any of these assumptions are weakened, the current regret guarantees would likely need significant changes.

**Strengths And Weaknesses:**

I think the paper addresses a real and interesting problem. Learning uplift in auctions while enforcing spending constraints is not easy, and it’s good to see a framework that tries to handle both at once.

The analysis is careful. In particular, the discussion about the dual variable under return-on-spend is thoughtful. The paper clearly explains that the multiplier can blow up and interact badly with learning error, and it proposes a concrete fix through a burn-in phase and a feasibility slack estimate. That part is not trivial, and I appreciate that the authors did not just hand-wave it away.

The way the market price distribution is estimated is also clear. The split-sample idea is constructive and avoids relying on non-implementable existence results. So from a theoretical point of view, the pieces are coherent.

That said, I do have some reservations.

First, the assumption that the highest competing bid is observed every round is quite strong. In some auction systems you do get this, but in others you only observe price when you win, or you get partial signals. The current regret guarantees really rely on full observation of market prices. So this is not just a small modeling detail — it’s doing a lot of work.

Second, the return-on-spend result depends on a strong feasibility slack condition. If the problem is close to the constraint boundary, the bound on the dual variable can become large. In that case, the regret guarantee may become loose. The burn-in phase helps, but it still assumes that there is meaningful slack to begin with.

Third, the linear structure and the overlap-type condition are standard in linear bandit work, but they are still assumptions. In practice, auction prices and uplift can be messy and heavy-tailed. So the guarantees should be read as holding in a fairly structured setting.

Overall, I would say the theory is internally consistent and careful under the stated assumptions. My hesitation is less about correctness and more about how restrictive some of the assumptions are, especially full market price feedback and strong feasibility slack.

---

> ### Author Rebuttal · Authors · 2026-03-31
>
> Thanks for providing invaluable feedback on our manuscript. We address your concerns as follows.
>
> **The model assumes that the highest competing bid is observed even when losing...**
>
> Thanks for pointing this out. We can generalize our regret proof to **one-sided feedback** where the competitor's bid is observed only when we lose ($b_t < m_t$), or to the complementary regime where it is observed only when we win. Due to symmetry, we assume the competitor's bid can be observed when $b_t < m_t$.
>
> For censored data, we can apply Tobit Maximum Likelihood Estimation (MLE) with the help of our Assumption 2.4 (Gaussian noise: $m_t = \phi_\star^\top x_t + \xi_t$, with $\xi_t \sim \mathcal{N}(0, \sigma_\star^2)$)
> In the revised manuscript, we will include a dedicated subsection detailing this extension. Below, we outline the two key components: the Tobit MLE procedure and the novel Parametric Bernstein Oracle Lemma.
>
> ### 1. The Procedure: Tobit Censored Regression
> Under one-sided feedback, the observation mechanism exactly mirrors a Tobit Type I censored regression model. Using Olsen’s (1978) reparameterization (letting $\rho = 1/\sigma_\star$ and $\theta = \phi_\star/\sigma_\star$), the Negative Log-Likelihood (NLL) for a single round $s$ is:
>
> $$ \ell_s(w) = - \mathbb{I}[b_s < m_s] \left( \log \rho - \frac{1}{2}\big(w^\top \tilde{x}_s(m_s)\big)^2 \right) - \mathbb{I}[b_s \ge m_s] \log \Phi\big(w^\top \tilde{x}_s(b_s)\big)$$
>
> where $w = [\theta^\top, \rho]^\top$ and $\tilde{x}_s(y) = [-x_s^\top, y]^\top$.
>
> This NLL is globally strictly convex. Because our bids $b_s$ are predictable (chosen based on the history $\mathcal{F}\_{s-1}$), the score function $\nabla \ell_s(w_\star)$ is an exact Martingale Difference Sequence (MDS). Standard self-normalized martingale concentration yields a parameter error of $\\|\hat{w}\_t - w\_{\star}\\|_{V_t} \le \epsilon_t = \tilde{\mathcal{O}}(d/\sqrt{t})$. We then analytically reconstruct the CDF as $\hat{F}_t(b) = \Phi\big(\hat{w}_t^\top \tilde{x}_t(b)\big)$.
>
> ### 2. The Key Lemma: Parametric Bernstein Oracle
> The following new lemma proves that the parametric Tobit estimator naturally yields the required Bernstein bound, smoothly shrinking the error in the tails.
>
> **Lemma (Parametric Tobit-Bernstein Oracle Bound):**
> *Let the reconstructed CDF be $\hat{F}\_t(b) = \Phi\big(\hat{w}\_t^\top \tilde{x}\_t(b)\big)$ and the true CDF be $F_t(b) = \Phi\big(w_\star^\top \tilde{x}_t(b)\big)$. For any $\delta \in (0,1)$, with high probability, the parameter error is bounded by $\epsilon_t = \tilde{\mathcal{O}}(d/\sqrt{t})$, and uniformly for all $b \in [0,1]$:*
> $$ |\hat{F}_t(b) - F_t(b)| \le \left(\sqrt{\frac{2}{\pi}} \epsilon_t\right) \sqrt{F_t(b)(1 - F_t(b))} + \left(\frac{1}{2\sqrt{2\pi e}}\right) \epsilon_t^2 $$
>
> **Proof Sketch:**
> Let $z = w\_\star^\top \tilde{x}\_t(b)$ and $\hat{z} = \hat{w}\_t^\top \tilde{x}\_t(b)$. From the MDS concentration, $|\hat{z} - z| \le \epsilon_t$. By Taylor expanding the standard normal CDF $\Phi$:
> $$ |\Phi(\hat{z}) - \Phi(z)| \le \phi(z)|\hat{z} - z| + \frac{1}{2} \left( \sup_\xi |\phi'(\xi)| \right) |\hat{z} - z|^2 \le \phi(z) \epsilon_t + \frac{1}{2\sqrt{2\pi e}} \epsilon_t^2 $$
> Then the proof is finished by considering the property of Gaussian distribution:
> $$ \phi(z) \le \sqrt{\frac{2}{\pi}} \sqrt{\Phi(z)(1-\Phi(z))} = \sqrt{\frac{2}{\pi}} \sqrt{F_t(b)(1-F_t(b))}. $$ $\blacksquare$
>
> We thank the reviewer again for prompting this extension, which significantly strengthens the practical relevance of our paper.
>
> **In the return-on-spend case, how does the regret bound behave as the feasibility slack gets small? Is the dependence explicit anywhere, or does it effectively hide inside constants?**
>
> In our manuscript, we have used $\Lambda$ as an approximation to the upper bound on the optimal Lagrangian multiplier. Based on Lemmas 5.2 and 5.4, we can conclude that $\Lambda=O(\frac{1}{\delta})$. By considering this relation $\Lambda=O(\frac{1}{\delta})$, we have explicitly highlighted the dependence on $\delta$ everywhere, and thus the regret guarantee is $\tilde{O}(\frac{d\sqrt{T}}{\delta})$. Interestingly, the expected net variation of the constraint does not rely on $\delta$.
>
> **The overlap-type condition ensures enough variation for stable estimation...**
>
> We can remove the sufficient overlap condition from Assumption 2.7 based on the following lemma, which can be considered as a corollary of Pontryagin's maximum principle.
> > **Lemma (Bounded Density Guarantees Overlap):**
> > *Suppose $\|f_t\|_\infty \le L$, $F_t(0)=0$ and $F_t(1)=1$. Under a uniform continuous bidding policy $b \sim U[0,1]$, the expected variance is strictly bounded away from zero:*
> > $$ \mathbb{E}\_{b \sim U[0,1]} [ F_t(b)(1 - F_t(b)) ] \ge \frac{1}{6L} \coloneqq c_{\text{var}} > 0 $$
>
> Intuition: Because $F_t$ is $L$-Lipschitz, the CDF cannot jump instantly from 0 to 1; it mathematically *must* spend "time" transitioning, meaning it is forced to have a strictly positive variance interval.

---

> > ### Author Rebuttal · Reviewer_9i55 · 2026-03-31
> >
> > Thank you for your detailed response. I appreciate the clarification and will keep my score as is.

---

### Official Review · Reviewer_8UA4 · 2026-03-11

**Soundness:** 3
**Presentation:** 3
**Significance:** 3
**Originality:** 3
**Overall Recommendation:** 4
**Confidence:** 2

**Summary:**

The paper studies the problem of online first price bidding under Linear Treatment Effect model, i.e.  the increase in value can be expressed as a linear model based on observable context vectors, depending on a unknown latent vector. The paper focus on the case in which not only the singular considered bidder's reward depend on the context but also the competition ( and therefore the bids of the competitors) is linearly effected by the context.

The paper study three different case on the base of the considered constraint: the unconstrained problem, the problem constrained by a maximum budget parameter, and the constraint on the Return-on-Spend of 1.

The paper a proposes a primal-dual framework that jointly learns the latent valuation parameters and the competitor’s bid distribution, proving under some conditions a regret of order $d\sqrt{T}$ for the unconstrained problem and for both type of constraints .

**Compliance With Llm Reviewing Policy:**

Affirmed.

**Final Justification:**

I have read the responses and keep the original score

**Key Questions For Authors:**

--

**Limitations:**

yes

**Strengths And Weaknesses:**

**Strength:** The problem studied by the paper is well motivated, in particular considering that the behavior of others bidders may depend on the same context of considered bidder seems well justified by practical applications. The paper is in general well written and quite understandable. The focus on providing efficient solutions seems of crucial importance considering the applications, like online advertising.

**Weakness:** While for at least part of the contributions the efficiency seems to be the key (since as the authors themselves claim there were preexisting alternative albeit less efficient) the complexity is never explicitly discussed or stated. A schematic comparison would be helpful.

On a smaller note, in the introduction, the authors claim *"Assuming linearity with respect to a feature vector does not imply the world is simple, but rather that the feature extractor is competent. "* This sentence might be misleading, as it ignores the fact that, while true that that a "complex" world might be well approximated by a linear dependence on a feature vector, restricting the dependence on the feature vector to a linear one may make the dimension of the feature vector $d$ explode, and considering that the guarantees have a linear dependence on $d$, this is quite relevant.

Minor: The assumptions seem to be cited all over the paper as theorem, like in theorem 2.8 *"Suppose Theorems 2.1 to 2.4, there exists a bidding algorithm..."*.

---

> ### Author Rebuttal · Authors · 2026-03-31
>
> We sincerely thank the reviewer for their encouraging feedback and strong support of our work. We address your questions as follows.
>
> **A schematic comparison (on the complexity) would be helpful.**
>
> We thank the reviewer for pointing out this interesting comparison. The procedure suggested in [1] relies on a partition guaranteed by the Marcus-Spielman-Srivastava (MSS) theorem (and the resolution of the Kadison-Singer problem). As shown in [2], this problem suffers from FNP-hardness. Next, a natural question is: why can our procedure bypass this hardness result?
> The apparent contradiction stems from the fact that the two methods are optimizing for fundamentally different mathematical guarantees. The computational difficulty of the MSS theorem arises from its strict, dimension-free requirements, which random coin-flipping cannot achieve. We summarize the core differences below:
> * Regularization vs. Raw Matrices: The random splitting lemma relies fundamentally on the regularization parameter $\lambda I$. As shown in the proof, random sampling fails to preserve the matrix structure in low-variance ("small signal") directions. The method bypasses this failure by artificially padding the matrix with $\lambda I$ to absorb the errors. In contrast, the MSS theorem applies to the *raw* sums of the vectors without any regularization padding. Partitioning unregularized matrices perfectly is a vastly harder combinatorial problem.
> * Dimensional Dependence vs. Dimension-Free: To guarantee concentration, random splitting requires $\lambda \ge 16 L^2 \log(d/\delta)$, meaning the approximation degrades as dimension $d$ grows. The breakthrough of MSS is its dimension-free bound, $(\frac{1}{\sqrt{r}} + \sqrt{\delta})^2$. Eliminating this $\log(d)$ penalty requires complex deterministic balancing (via interlacing polynomials) rather than standard matrix concentration inequalities.
> * One-Sided Bounds vs. Tight Two-Sided Bounds: The random splitting lemma guarantees a loose, one-sided lower bound with high probability. The MSS theorem guarantees an incredibly tight, deterministic upper bound for *every single subset* in the partition simultaneously.
>
> In summary, the random splitting lemma demonstrates that finding a loose, regularized, dimension-dependent approximation is computationally easy (achievable via random sampling). However, finding the tight, unregularized, dimension-independent exact partition guaranteed by MSS requires solving a highly constrained combinatorial puzzle, which recent literature has proven to be FNP-hard. We will add a brief note after Lemma 3.2 to clarify this distinction for future readers.
>
> **On a smaller note, in the introduction, the authors claim "Assuming linearity with respect to a feature vector does not imply the world is simple, but rather that the feature extractor is competent." This sentence might be misleading...**
>
> We sincerely thank the reviewer for this insightful correction. You are absolutely right: forcing linearity via simple feature expansion would inflate $d$ and render our $\tilde{O}(d\sqrt{T})$ regret bound practically vacuous. We agree our original phrasing failed to convey this crucial tension between representation power and dimensionality.
>
> Our intent was to refer to the standard architecture of modern ad pipelines. Rather than expanding features, platforms use deep representation learning to compress complex raw inputs into **compact, low-dimensional embeddings** (e.g., $d = 64$ or $128$). A "competent" feature extractor is precisely one that maps complex data into a space where linear approximations work *without* inflating $d$.
>
> To ensure mathematical rigor, we will replace the misleading sentence in our introduction with the following clarification:
>
> > *"While the underlying auction environment is highly complex, modern ad platforms typically rely on upstream representation learning to map raw, high-dimensional data into compact, low-dimensional embeddings. Following standard abstractions in the linear bandit literature, we assume that the value uplift and market price can be reasonably approximated by linear functions over these compact context vectors. This assumption balances modeling expressivity with the need to keep the feature dimension $d$ small, which is critical since our regret guarantees exhibit a necessary linear dependence on $d$."*
>
> **The assumptions seem to be cited all over the paper as theorems.**
>
> Thanks to the reviewer for pointing this out. This phenomenon is due to abnormal behavior of the `cref` package. We promise to fix this in the revised version.
>
> [1] Wen, Yuxiao, Yanjun Han, and Zhengyuan Zhou. "Joint value estimation and bidding in repeated first-price auctions." arXiv preprint arXiv:2502.17292 (2025).
>
> [2] Jourdan, Ben, Peter Macgregor, and He Sun. "Is the algorithmic Kadison-Singer problem hard?." arXiv preprint arXiv:2205.02161 (2022).

---

> > ### Author Rebuttal · Reviewer_8UA4 · 2026-04-04
> >
> > I would like to thank the reviewer for their detailed answer and I am inclined to keep my original score.

---

### Official Review · Reviewer_NnjA · 2026-03-13

**Soundness:** 2
**Presentation:** 2
**Significance:** 3
**Originality:** 2
**Overall Recommendation:** 3
**Confidence:** 3

**Summary:**

This paper studies online bidding in repeated first-price auctions when an advertiser’s impression value is latent and must be inferred from observational data, and when global financial constraints (hard budget or expected RoS) are present. The authors posit a Linear Treatment Effect (LTE) model in which both the uplift value and the highest competing bid depend linearly on the same context vector, and they develop a unified primal–dual framework that learns the valuation and the competitor distribution jointly.

**Compliance With Llm Reviewing Policy:**

Affirmed.

**Final Justification:**

I tend to choose `weak reject`.

**Key Questions For Authors:**

- How computationally intensive is the convexification step in PLANROS, and what is the concrete implementation of “Deng’s transformation”? Can you provide complexity or numerical stability considerations for large K or continuous bids?

- The model assumes full-information feedback on $m_t$. How would your estimation and regret guarantees degrade under one-sided or censored feedback (observe $m_t$ only when losing or only winning price)? Could the split-sample idea be adapted in that case?

**Limitations:**

Yes.

**Strengths And Weaknesses:**

## Strength

- This paper introduces a “doubly linear” setting where both treatment uplift and opponents’ highest bid depend on the same context, filling a gap between prior works that treated only one side as linear.

- This paper proposes a constructive split-sample CDF estimator grounded in a new spectral approximation lemma, providing a practical alternative to non-constructive subset selection used in earlier LTE analyses.

- The theoretical analysis of the paper is clearly structured with regret decomposition and uses standard self-normalized concentration with elliptical potential tools.

## Weakness

- **Assumptions are heavy**: i.i.d. contexts, Gaussian noise on bids, bounded density/Lipschitz CDF, unconfoundedness, strong Slater with margin $\delta$, and an overlap condition. The analysis’ dependence on these assumptions may limit robustness.

- **Presentation Issues**: Several notational/numbering inconsistencies (e.g., “Theorems 2.1–2.4” appear to be assumptions; small artifacts in equations/algorithms) make verification harder than necessary.

---

> ### Author Rebuttal · Authors · 2026-03-31
>
> **Computational cost of the convexification step in PLANROS; the complexity or numerical stability for large $K$ or continuous bids:**
>
> We sincerely thank the reviewer for this highly practical question. To ensure scalability for large grid sizes $K$ and continuous domains, our implementation of `PLANROS` executes both Deng’s transformation and the UCB grid search in **$O(\log K)$ time per step**, completely removing the computational bottleneck.
>
> Here are our implementation details and complexity guarantees:
>
> **1. Deng’s Transformation via Dynamic Convex Hull ($O(\log K)$ update)**
> - The Naive Implementation: For each bid $b \in \mathcal{B}_K$, we map it to a 2D point $(x, y) = (b \cdot F^\dagger(b), F^\dagger(b))$, representing expected payment and allocation. We sort these $K$ points by their $x$-coordinate and run a standard convex hull algorithm (e.g., Andrew's Monotone Chain) to extract the upper envelope. Rebuilding this from scratch at every round $t$ takes $O(K \log K)$ time.
>
> - Optimized Incremental Update ($O(\log K)$): Because the empirical CDF $\hat{F}_t(b)$ receives exactly one new observation per round, the update to the points is highly localized. Instead of rebuilding from scratch, we maintain the base empirical envelope $(b \cdot \hat{F}_t(b), \hat{F}_t(b))$ using a Dynamic Convex Hull data structure (e.g., a balanced Binary Search Tree). This allows us to insert or update the local point in strictly $O(\log K)$ time.
> * Lazy $\epsilon_t$ Shift: Because the optimistic radius $\epsilon_t$ shrinks globally, adding it explicitly would trigger $O(K)$ updates. Instead, we maintain the hull purely on empirical data and treat $\epsilon_t$ as a "lazy" affine shift, applying it at $O(1)$ cost only when querying the envelope.
>
> **2. UCB Grid Search via Convex Optimization ($O(\log K)$ search)**
> We bypass the need for an $O(K)$ linear scan over all candidate bids.
> * The Concavity Property: The UCB estimators ($u_{t,0}$ and $u_{t,1}$) are strictly affine functions of expected payment and allocation. Because Deng's transformation guarantees a concave allocation curve, our final Lagrangian objective $J(x)$ is strictly concave over the sorted 1D payment space.
> * Fast Search: Since $J(x)$ is unimodal along the envelope, we find the optimal bid in $O(\log K)$ time using a Binary Search over the marginal slopes of the hull segments.
>
> **3. Continuous Domains and Numerical Stability**
> * Continuous Regret vs. Complexity: For continuous spaces, we discretize into a grid of size $K = \sqrt{T}$. The $O(T/K)$ discretization error perfectly absorbs into our $O(\sqrt{T})$ regret bound. While a growing grid normally causes a severe computational bottleneck, our search bounds the per-step complexity to $O(\log T)$, achieving the theoretically optimal continuous regret without sacrificing real-time feasibility.
> * Numerical Regularization: Because $K = \sqrt{T}$ becomes exceptionally large for long time horizons, the discrete grid $\mathcal{B}\_K$ acts as a vital numerical regularizer. To prevent floating-point instability (e.g., collinearity during dynamic hull updates) when points are highly dense, we use a standard $\epsilon_{\text{tol}}$ tolerance for cross-product checks.
>
> In summary, the per-step complexity of the convexified oracle in `PLANROS` is strictly bounded by $O(\log K)$, making it highly stable. We will include a discussion detailing these data structure optimizations in the revised manuscript.
>
> **How would your estimation and regret guarantees degrade under one-sided or censored feedback?**
>
> Please see the response to Reviewer 9i55 on "The model assumes that the highest competing bid is observed even when losing..." for details.
>
> **Assumptions are heavy:**
>
> Thanks for pointing this out. After careful examination, we find that we can eliminate Assumption 2.3 (Bounded Density) and the Sufficient Overlap Condition in Assumption 2.7. We illustrate how to eliminate Assumption 2.3 as follows, and we defer how to eliminate the sufficient overlap condition to response to Reviewer 9i55. Based on Assumption 2.4, the competitor's bid is generated as $m_t = \phi_\star^\top x_t + \xi_t$, where $\xi_t \sim \mathcal{N}(0, \sigma_\star^2)$. Let $Q(v)$ be the CDF of this Gaussian noise. The CDF of the market at a continuous bid $b$ is exactly:
> $$F_t(b) = \mathbb{P}(\phi_\star^\top x_t + \xi_t \le b) = Q(b - \phi_\star^\top x_t)$$
> The Lipschitz constant of $F_t$ is the global supremum of its derivative with respect to $b$. Since $F_t'(b) = Q'(b - \phi_\star^\top x_t)$, the Lipschitz constant is simply the peak of the Gaussian probability density function:
> $$\sup_{b \in \mathbb{R}} |F_t'(b)| = \sup_{v \in \mathbb{R}} Q'(v) = \frac{1}{\sigma_\star\sqrt{2\pi}}.$$
>
> **Presentation Issues:**
>
> Thanks for pointing this out. The assumptions are incorrectly labeled as theorems due to abnormal behavior of the `cref` package. We promise to fix these presentation issues in the revised version.

---

> > ### Author Rebuttal · Reviewer_NnjA · 2026-04-03
> >
> > Thank you for your response. I will raise my score by 1.

---

### Decision · Program_Chairs · 2026-04-30

**Decision:**

Reject

**Comment:**

While reviewers recognized that the paper presents a well motivated framework, they raised concerns regarding the restrictiveness of its core theoretical assumptions. Specifically, the reviewers have concern about heavy reliance on strong assumptions. While the authors generalized from full feedback to one sided feedback during the rebuttal, reviewers expressed significant reservations about the strong Slater condition. Even after the authors provided justifications during the rebuttal, concerns remained that the algorithm's performance and regret guarantees are highly dependent on this substantial feasibility slack.

The rebuttals and the authors' proposed theoretical extensions were carefully reviewed. However, the consensus remains that the restrictive nature of the assumptions outweighs the conceptual merits. Consequently, the paper is recommended for rejection.